# Coactivators and general transcription factors have two distinct dynamic populations dependent on transcription

Nikolaos Vosnakis[1,2,3,4], Marc Koch[1,2,3,4] (iD), Elisabeth Scheer[1,2,3,4], Pascal Kessler[1,2,3,4] (iD), Yves Mély[4,5], Pascal Didier[4,5] & László Tora[1,2,3,4,*] (iD)

## Abstract

SAGA and ATAC are two distinct chromatin modifying co-activator complexes with distinct enzymatic activities involved in RNA polymerase II (Pol II) transcription regulation. To investigate the mobility of co-activator complexes and general transcription factors in live-cell nuclei, we performed imaging experiments based on photobleaching. SAGA and ATAC, but also two general transcription factors (TFIID and TFIIB), were highly dynamic, exhibiting mainly transient associations with chromatin, contrary to Pol II, which formed more stable chromatin interactions. Fluorescence correlation spectroscopy analyses revealed that the mobile pool of the two co-activators, as well as that of TFIID and TFIIB, can be subdivided into "fast" (free) and "slow" (chromatin-interacting) populations. Inhibiting transcription elongation decreased H3K4 trimethylation and reduced the "slow" population of SAGA, ATAC, TFIIB and TFIID. In addition, inhibiting histone H3K4 trimethylation also reduced the "slow" populations of SAGA and ATAC. Thus, our results demonstrate that in the nuclei of live cells the equilibrium between fast and slow population of SAGA or ATAC complexes is regulated by active transcription via changes in the abundance of H3K4me3 on chromatin.

Keywords  diffusion constant; FCS; FLIP; FRAP; tudor domain
Subject Categories  Chromatin, Epigenetics, Genomics & Functional Genomics; Transcription
The EMBO Journal (2017) 36: 2710–2725

## Introduction

SAGA (Spt-Ada-Gcn5 acetyltransferase) is an evolutionary conserved, multifunctional co-activator complex with modular organization (Spedale *et al*, 2012). SAGA contains about 18-20 subunits, which are organized in histone acetyltransferase (HAT), histone H2Bub1 deubiquitinase (DUB) and TATA binding protein (TBP) regulatory and structural modules (Lee *et al*, 2011; Spedale *et al*, 2012). Metazoan GCN5 was also identified as a subunit of a second coactivator HAT complex in *Drosophila* and mammals, named ATAC (Ada Two A Containing; Guelman *et al*, 2006; Wang *et al*, 2008; Nagy *et al*, 2010). ATAC contains 10 well-characterized subunits. Three subunits of the HAT modules, GCN5 (or PCAF), SGF29 and ADA3, are shared between SAGA and ATAC complexes. Several SAGA and ATAC subunits have "reader" domains that interact directly with chromatin by recognizing specific transcription-associated histone modifications. For example, GCN5 (or PCAF) has a bromodomain that binds preferentially to acetylated H3 and H4 histone tail peptides (Dhalluin *et al*, 1999; Hudson *et al*, 2000), while SGF29 contains tandem tudor domains, which was shown to bind H3K4me3 (Vermeulen *et al*, 2010; Bian *et al*, 2011).

The differences in subunit composition between the two distinct HAT complexes have been suggested to play regulatory roles for SAGA and ATAC (Nagy *et al*, 2009, 2010). Genome-wide association studies using ChIP-seq against specific subunits of either SAGA or ATAC in different organisms revealed only a few hundred binding sites for these complexes (Vermeulen *et al*, 2010; Bian *et al*, 2011; Krebs *et al*, 2011; Venters *et al*, 2011; Weake *et al*, 2011). In most cases, SAGA subunits were detected at a subset of promoters and at gene bodies of specific genes, while the overlap between SAGA and ATAC at mammalian promoters was limited (Govind *et al*, 2007; Krebs *et al*, 2011; Venters *et al*, 2011). Those findings together with the fact that in yeast there was only a very weak overlap between SAGA-bound and SAGA-regulated genes, when analysed at the steady state mRNA level, raised questions about the detectability of these coactivators by ChIP (Lenstra & Holstege, 2012 and refs therein). Indeed, when the genome-wide distribution of the SAGA co-activator complex was tracked by following its two chromatin modifying activities (H3K9 acetylation or H2Bub1 deubiquitination),

1  Institut de Génétique et de Biologie Moléculaire et Cellulaire, Illkirch, France
2  Centre National de la Recherche Scientifique, UMR7104, Illkirch, France
3  Institut National de la Santé et de la Recherche Médicale, U964, Illkirch, France
4  Université de Strasbourg, Illkirch, France
5  Laboratoire de Biophotonique et Pharmacologie, Illkirch, France
   *Corresponding author. Tel: +33 3886 53444; E-mail: laszlo@igbmc.fr

  

it was discovered that SAGA acetylates the promoters, and deubiquitinates the transcribed regions, of all expressed genes both in *S. cerevisiae* and in human cells (Bonnet *et al*, 2014). The latter study provided data suggesting that SAGA DUB interacts with chromatin in a Pol II transcription-independent manner. Studies in different eukaryotes have suggested that the presence of SAGA at the promoters of genes is needed to facilitate Pol II recruitment and pre-initiation complex (PIC) formation (Wyce *et al*, 2007; Nagy *et al*, 2009; Helmlinger *et al*, 2011; Lang *et al*, 2011). The recruitment of SAGA and ATAC coactivator complexes to genomic loci has been suggested to take place by several distinct mechanisms: (i) by activator mediated recruitment (McMahon *et al*, 1998; Brown *et al*, 2001; Fishburn *et al*, 2005; Reeves & Hahn, 2005), (ii) by interactions with basal transcription machinery components (Larschan & Winston, 2001; Laprade *et al*, 2007; Mohibullah & Hahn, 2008) and (iii) through chromatin-interacting domains of SAGA and ATAC subunits (Hassan *et al*, 2002; Vermeulen *et al*, 2010; Bian *et al*, 2011; Bonnet *et al*, 2014). Nevertheless, the dynamics of SAGA and ATAC interactions with chromatin are not yet well understood, as there has been no direct and systematic monitoring of the nuclear mobility of these two co-activator complexes in live cells.

Through the application of various microscopy-based techniques (Day & Schaufele, 2008), it has been well established that the function of proteins (or proteins complexes) involved in chromatin-dependent processes is related to their mobility and their interactions with the nuclear environment (Kimura *et al*, 1999, 2002; Dundr *et al*, 2002; Kimura, 2005; Gorski *et al*, 2008; van Royen *et al*, 2011). Our current understanding of transcription regulation dynamics has been expanded by the application of imaging-based live-cell approaches, in which the equilibrium of fluorescently tagged factors within a region of interest (ROI) or a cellular compartment is altered, and subsequently, fluorescence redistribution is followed over time. Approaches like FRAP (Fluorescence Recovery After Photobleaching) and FLIP (Florescence Loss In Photobleaching), based on a single or repeated photobleaching of a ROI, respectively, have been successfully applied to characterize the interactions of a plethora of factors with chromatin in live cells and obtain information on their dynamics on a millisecond time scale (Snapp *et al*, 2003; Hager *et al*, 2009). Their application has been particularly advantageous in detecting the presence of free, transiently and, importantly, fully immobilized fractions of a given factor (Kimura, 2005; van Royen *et al*, 2009) in a 50 ms to a few minutes time window. Nevertheless, the potential of such techniques for analysing multiple diffusing subpopulations can be limited by the fact that only the average fluorescence intensity of the given ROI, or cellular compartment (e.g. nucleus), is monitored over time. This is important when the mobility parameters (i.e. diffusion) of multiple components of a dynamic factor need to be characterized. Fluorescence correlation spectroscopy (FCS) is a fluorescence microscopy technique based on the detection and quantification of fluctuations of fluorescence intensity of molecules diffusing through a well-defined (subfemtoliter) observation volume (Machan & Wohland, 2014). FCS is particularly useful for the analysis of diffusion properties of very mobile factors and can be used to gain more detailed information than those obtained by photobleaching approaches. The single-molecule sensitivity and the microsecond to a few seconds time resolution of FCS have for example

allowed the dynamic repartitioning of DNA-binding factors driven by physiological epigenetic changes, provided quantitative insights into chromatin-binding dynamics of the Polycomb complex or participated to characterizing protein network dynamics in live cells (Steffen *et al*, 2013; Wachsmuth *et al*, 2015; White *et al*, 2016). Fluorescence intensity correlation analyses describe several physicochemical parameters of labelled factors, such as the total number of diffusing molecules ($N_{tot}$), their apparent diffusion coefficient ($D$), and their apparent molecular weight ($MW_A$). Importantly, the apparent diffusion coefficient provides an objective reference value to compare mobility properties between different factors.

To obtain information on the mobility of human SAGA/ATAC co-activator complexes at the subsecond range, and to investigate the presence of their immobile fractions, we performed FRAP and FLIP. Our measurements indicate that, in contrast to RPB1 (Pol II subunit), SAGA and ATAC subunits are highly dynamic and exhibit only transient interactions with chromatin with no detectable immobile fractions. Similar mobility was also found for the general Pol II transcription factors, TAF5 (TFIID subunit) and TFIIB. To characterize the diffusion of TFIIB, TFIID, SAGA, and ATAC complexes in the nuclei of living cells with high sensitivity, we performed live-cell FCS experiments to study those factors. Our FCS analyses show that two distinct populations dominate the diffusion of coactivators (SAGA and ATAC) and GTFs (TFIIB and TFIID): a "fast" population (having the approximate size of the studied complexes) and a "slow", probably chromatin-interacting population (with a much higher $MW_A$). Importantly, the slow fraction of SAGA and ATAC is reduced upon inhibition of transcription elongation or histone H3K4 trimethylation suggesting that association of SAGA and ATAC with chromatin in live cells is largely mediated by changes in the levels of transcription and the associated H3K4me3 histone mark.

# Results

## Real-time chromatin association of SAGA and ATAC is comparable to highly dynamic PIC components, TAF5 and TFIIB, but different from Pol II

To analyse the mobility of SAGA and ATAC coactivators in the nuclei of living cells and to compare their dynamics with other key players of Pol II transcription machinery, we first used FRAP. To visualize subunits of transcription factors in living cells, we used an approach based on genetically encoded fluorescence labelling, whereby the coding sequences of several subunits of the different complexes were cloned in frame with eGFP (Materials and Methods) and the obtained plasmids were transiently transfected in human U2OS cells. All the generated plasmids expressed the fusion proteins with the expected sizes (Appendix Fig S1A) and immunoprecipitation experiments indicated that in contrast to unconjugated eGFP (Appendix Fig S1B), the eGFP-tagged subunits could incorporate into the corresponding complexes (Appendix Fig S1C–G). To ensure the incorporation of eGFP-tagged proteins in the corresponding complexes for imaging experiments, they were weakly expressed. To be able to detect low levels of fluorescence signal in single cells, an EMCCD (Electron Multiplying Charge Coupled Device) camera

was used. Moreover, the individual cells selected for photobleaching experiments expressed the different eGFP-tagged proteins at similar levels (Appendix Fig S2).

First, we chose RPB1 (the largest subunit of Pol II) known to have stable interactions with chromatin (Kimura *et al*, 2002; Hieda *et al*, 2005; Boireau *et al*, 2007; Darzacq *et al*, 2007; Fromaget & Cook, 2007) and also highly dynamic nuclear factors, like the TFIID subunit TAF5 and the GTF TFIIB (Chen *et al*, 2002; Sprouse *et al*, 2008; de Graaf *et al*, 2010; Ihalainen *et al*, 2012). In addition, eGFP was used as a control for a freely diffusing protein that should exhibit no specific interactions with the nuclear environment. The raw fluorescence intensity in the ROI of the an eGFP-expressing cell was quickly increased within 1 s after the bleach (Fig 1A). The FRAP ROI of the GCN5-expressing cell reached full recovery after the first 10 s (Fig 1A), whereas raw fluorescence intensity in the ROI of the RPB1-expressing cell did not (Fig EV1A). FRAP kinetics of eGFP, RPB1, TAF5 and TFIIB were in good agreement with previously published data (Kimura *et al*, 2002; Hieda *et al*, 2005; de Graaf *et al*, 2010; Fig EV1B–D).

Next, we used FRAP to compare for the first time the nuclear dynamics of common (GCN5, SGF29; Fig 1B and C), SAGA (SPT20, USP22; Fig 1D and E), and ATAC-specific (ZZZ3; Fig 1F) subunits in live cells. We observed that for every tested subunit fluorescence recovery after the bleach occurred within the order of 10 seconds with no detectable immobile fraction (as suggested by the full recovery of FRAP curves; Fig 1B–F and Appendix Fig S3). Moreover, our FRAP measurements indicated that all SAGA and ATAC subunits show very similar recovery kinetics to each other (Fig 1G), and to the highly mobile factor, TAF5 (Fig 1H), which is known to interact only transiently with chromatin (de Graaf *et al*, 2010; Ihalainen *et al*, 2012). In contrast to the highly mobile behaviour of SAGA, ATAC, and TFIID subunits, fluorescence recovery of RPB1 occurred at a slower rate (Fig 1I and J).

The rapid and complete recovery of fluorescence observed for SAGA and ATAC subunits suggests that these factors associate in a very dynamic manner with chromatin without detectable immobile fraction. To exclude the possibility that the small size of ROI used in FRAP combined with low signal-to-noise ratio (S/N) might influence the FRAP measurements, we complemented our live-cell analyses by performing FLIP. Importantly, the exposure time in FLIP is 500 ms, 10 times longer than that used for FRAP (50 ms), allowing the selection of cells that express the given GFP-tagged factors at even lower levels than in FRAP, with a higher S/N ratio.

A comparison between representative eGFP-NLS-, GCN5- and RPB1-expressing cells subjected to FLIP is shown in Fig EV2A. Raw nuclear fluorescence intensity of the eGFP-NLS-expressing cell was dramatically decreased after 30 s (10 bleach pulses). Signal in the nucleus of GCN5-expressing cell was still visible after 60 s (20 pulses), whereas nuclear intensity of the RPB1-expressing cell was reduced at much slower rate with the signal remaining visible even after 240 s (80 pulses). Normalized FLIP curves showed that the depletion of fluorescence in RPB1-expressing cells was not complete even after 4 min (Fig EV2B), in agreement with previous studies (Darzacq *et al*, 2007; Fromaget & Cook, 2007). On the contrary, TAF5 and TFIIB FLIP curves showed a rapid loss of fluorescence intensity, which reached background levels after 4 min (Fig EV2C, D and M). FLIP analyses of SAGA and ATAC subunits

further demonstrated the absence of significant immobile fraction (Fig EV2E–I) and the similar mobility between SPT20/SAGA and ZZZ3/ATAC complex-specific subunits (Fig EV2G, I and J). The comparison of GCN5 (shared SAGA/ATAC subunit) and TAF5 showed also that these co-activator complexes are as dynamic as the tested GTFs (Fig EV2K). In contrast, comparison of GCN5 with RPB1 dynamics (Fig EV2L), even at the whole nucleus scale, further demonstrated that SAGA and ATAC have very different behaviour in the nucleus when compared to Pol II. Thus, our FLIP and FRAP analyses together show that SAGA and ATAC are very dynamic with no detectable chromatin bound (immobile) fractions.

To test the above live-cell imaging findings with an independent biochemical approach, we performed subcellular fractionation of human cells and verified the relative abundance of endogenous SAGA, ATAC and Pol II, subunits in the different cellular and nuclear compartments (Fig 2). In agreement with our imaging experiments, we found that SAGA and ATAC subunits (common or complex-specific) are predominantly present in the nuclear soluble fraction. On the contrary, transcribing Pol II, as exemplified by Ser2 phosphorylated C-terminal domain (CTD) of RPB1, was mainly detected in the chromatin bound fraction (Fig 2; Hsin *et al*, 2014). Thus, the absence of significant amounts of SAGA and ATAC subunits in the chromatin bound fraction supports our imaging data.

### FRAP and FLIP analyses detect no global changes in SAGA and ATAC dynamics upon transcription inhibition

Our FRAP and FLIP analyses strongly suggest that TFIIB, TFIID, SAGA and ATAC complexes have different dynamics compared to Pol II. However, it remained unclear to what extent, if any, active transcription may affect the dynamics of the complexes. To answer this question, we applied FRAP and FLIP measurements in cells treated with flavopiridol (FVP). FVP is a potent cyclin-dependent kinase (CDK) inhibitor, which inhibits Pol II transcription elongation through the positive transcription elongation factor b (P-TEFb; Chao *et al*, 2000). To test the effect of FVP in transcription inhibition, we tracked changes in the phosphorylation levels of the CTD of RPB1 by Western blot analysis. After 1 h of treatment of the cells with 2 μM FVP, phosphorylation levels of the CTD decreased dramatically, indicating the disappearance of Pol II elongating fraction (Fig 3A). Cells were transfected with the plasmids encoding the respective eGFP fusion proteins; 16–24 h later, cells were treated with FVP (2 μM) and were subsequently subjected to FRAP or FLIP measurements 90–120 min following the treatment. Importantly, cells selected for measurements in control or transcription elongation inhibition conditions were expressing the eGFP-tagged proteins at similar levels (Appendix Fig S2). We first tested whether FVP had an effect on the mobility of free eGFP due to potential indirect structural changes in the nuclear environment. The comparison of average normalized FRAP curves of eGFP between flavopiridol-treated and control cells indicated that FVP does not influence the nuclear mobility of eGFP (Fig EV1E). Next, we tested the effect of FVP treatment on RPB1 dynamics and found a dramatic increase in fluorescence recovery compared to control cells (Fig EV1F) in agreement with previous studies (Fromaget & Cook, 2007; Missra & Gilmour, 2010). In contrast to RPB1, TAF5

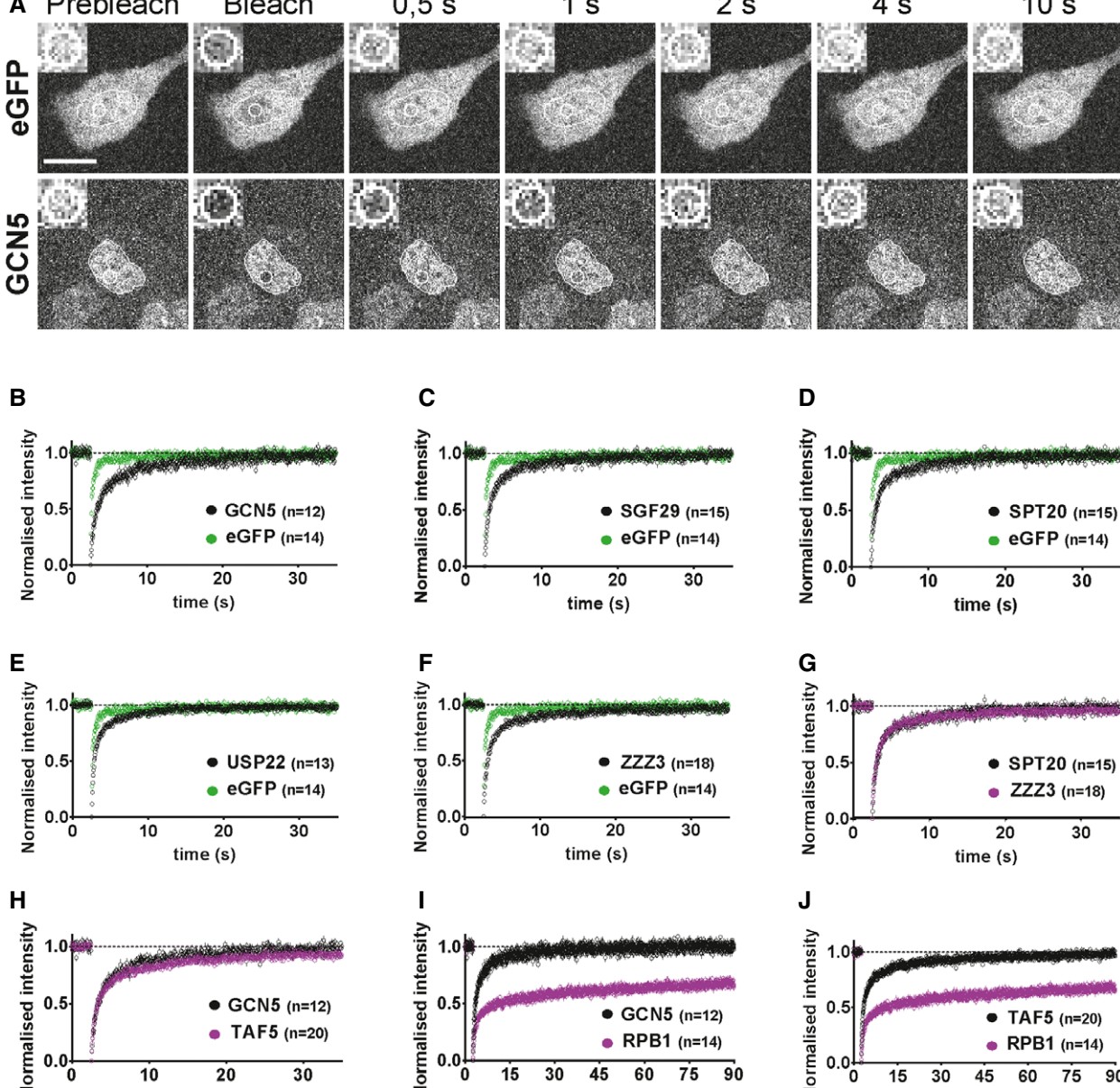

**Figure 1. FRAP analyses suggest that SAGA and ATAC subunits and GTFs, TFIIB and TAF5, are highly mobile in live-cell nuclei.**

A    FRAP experiments performed on eGFP- and eGFP-GCN5-expressing human U2OS cells. Representative images of the photobleached nuclear regions are shown. Time after photobleaching is indicated on top of the panels. An inset, showing a four times zoom-in of the bleached ROI (white circle) is shown at the top left corner of each frame. The nucleus of each cell is encircled with a white line. Scale bar is 20 μm.

B–F    Average normalized FRAP curves using the full-scale method (Ellenberg *et al*, 1997) were calculated, and eGFP FRAP curve was compared to (B) GCN5 and (C) SGF29 (shared SAGA/ATAC subunits; (D) SPT20 and (E) USP22 (SAGA subunits); (F) ZZZ3 (ATAC subunit). Only the first 35 s of recovery are shown to resolve better the initial time points which are more informative for proteins that recover fast.

G–J    Comparison of FRAP curves between (G) SAGA subunit, SPT20, and ATAC subunit, ZZZ3; (H) TAF5 (TFIID subunit) and GCN5 (SAGA/ATAC subunit); (I) GCN5 (SAGA/ATAC subunit) and RPB1 (Pol II subunit); and (J) TAF5 (TFIID subunit) and RPB1 (Pol II subunit). In the graphs (I, and J), the post-bleach time is extended to 90 s to better illustrate the differences between the recoveries of distinct factors having very different mobility.

Data information: On the *x*-axis, time is represented in seconds (s). *n*: number of cell nuclei analysed for each factor. Hereafter, in the text and in each figure panel, the names of the tested factors are indicated without including the eGFP moiety.

and TFIIB FRAP kinetics are not affected by transcription inhibition (Fig 3B and C). Similarly, to the behaviour of these highly mobile factors, the comparison of FRAP curves of SAGA and ATAC subunits obtained from control and FVP-treated cells showed no detectable changes (Fig 3D–H). These FRAP measurements suggested that inhibition of transcription elongation does not

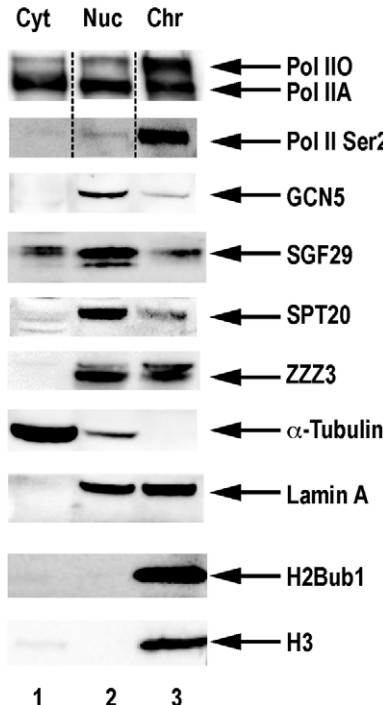

**Figure 2. Subcellular distribution of endogenous SAGA and ATAC subunits in human cells.**

Western blot analysis of the abundance of shared, and SAGA- or ATAC-specific subunits in the cytoplasmic (Cyt), nuclear soluble (Nuc) and chromatin-associated (Chr) fractions of human HEK 293 cells. Equivalent amounts of protein from each fraction (20 μg) were loaded as indicated. The fractions were tested with antibodies raised against either the non-phosphorylated form of CTD of RPB1 subunit of Pol II, or raised against the Ser2 phosphorylated form of CTD (Ser2-Pol II). Pol IIA and Pol IIO forms of RPB1, that correspond to hypo- (IIA) and hyper-phosphorylated (IIO) CTDs of RBP1, respectively, are labelled. Distribution of SAGA and ATAC was analysed by probing the blots with antibodies raised against complex-specific, SPT20 (SAGA) and ZZZ3 (ATAC), and shared, GCN5 and SGF29, subunits. Antibodies raised against α-tubulin and histones (H3 or H2Bub1) were used as markers of the cytoplasmic or chromatin fractions, respectively. An antibody raised against lamin A was used as a common marker of nuclear soluble and chromatin fraction. Dotted lines indicate that the blots were cut.

Source data are available online for this figure.

significantly modify the mobility of SAGA and ATAC, as well as that of TAF5 and TFIIB.

To investigate whether monitoring changes at the whole nucleus scale could provide additional information about the mobility of the tested factors during transcription elongation, we also applied FLIP on FVP-treated cells. As for FRAP, after FVP treatment we detected the dramatic dissociation of Pol II from chromatin (Fig EV3A), in agreement with previous observations (Kimura *et al*, 2002; Hieda *et al*, 2005). However, both in the case of the highly mobile PIC components, TAF5 and TFIIB (Fig EV3B and C) and the tested SAGA and ATAC subunits (Fig EV3D–H), FLIP analyses detected no, or only minor deviations, from the average curves generated from non-treated cells. Thus, by using two different photobleaching techniques, we observed no detectable changes in SAGA and ATAC, as well as TFIID and TFIIB, dynamics upon transcription inhibition.

## Live-cell FCS reveals that the diffusion of SAGA and ATAC and PIC components, TAF5 and TFIIB, is dominated by the presence of two distinct populations

FRAP and FLIP suggest that SAGA and ATAC are factors which interact with chromatin in a very dynamic manner; however, these averaging techniques may not sufficiently resolve the diffusion differences of multiple, highly mobile components. We thus further assessed their mobility by FCS. To resolve fluorescence fluctuations at the single-molecule level, we derived our measurements from very small (0.4 fl) observation volumes, where the total number of molecules ($N_{tot}$) was lower than 200 (Materials and Methods and Appendix Fig S4).

As a control, we measured the diffusion of purified eGFP in aqueous buffer (eGFPaq). As expected, the average autocorrelation curve fits well with a one-component free diffusion model (Fig 4A). The single peak of the frequency distribution of diffusion constants, generated by the Maximum Entropy approach (Materials and Methods), was centred around 92 $\mu m^2$/s (Fig 4C), as previously described (Potma *et al*, 2001). Next, we analysed transiently expressed eGFP in living cells, whose average autocorrelation curve also fitted a one-component free diffusion model (Fig 4B), as expected for a protein with no specific interactions with the nuclear environment. The peak of the frequency distribution of the eGFP apparent diffusion constant centred around 31 $\mu m^2$/s (Fig 4C), in good agreement with previous studies (Bancaud *et al*, 2009; Kloster-Landsberg *et al*, 2012).

Interestingly, for all the other factors used in this study, a high confidence fit required two components in the free diffusion model (a representative fit, of GCN5, is shown in Fig 4D), suggesting that these particles can be classified into two populations. Moreover, the validity of the two populations model is supported by (i) the maximum entropy approach for which there is no *a priori* about the number of diffusing species and (ii) the low reduced chi square values obtained for all factors with the two-population fit (Appendix Table S1). Note that the addition of a third component did not increase the goodness of the fit (Fig 4D). Consistently, the unbiased fitting of FCS autocorrelation curves gave bi-modal distribution (two peaks) of apparent diffusion constants for common (GCN5, SGF29), SAGA (SPT20)- and ATAC (ZZZ3, YEATS2)-specific subunits, as well as the GTFs, TFIID/TAF5 and TFIIB (Fig 5A–G). Thus, these highly mobile proteins (as characterized by averaging photobleaching) can be further dissected into two populations: a *fast* one (with higher mean $D$, or $D_{fast}$) and a *slow* one (with lower mean $D$, or $D_{slow}$). These apparent $D_{fast}$ and $D_{slow}$ measures of individual SAGA or ATAC subunits showed comparable distributions to each other, as well as to the studied GTFs (Fig 5H).

We were able to indirectly infer the nature of these different populations by calculating their apparent molecular weights ($MW_A$) by comparing each factor's apparent $D_{fast}$ and $D_{slow}$ values with $D_{eGFP}$ (Materials and Methods). The *fast* components of the tested SAGA and ATAC subunits all had $MW_A$ values in the MDa range (Table 1), suggesting that the GFP-fused factors incorporated into endogenous SAGA and ATAC complexes, whose biochemical purifications have similar sizes (2 and 0.7 MDa, respectively; Wang *et al*, 2008; Nagy *et al*. 2010). Thus, these subpopulations of SAGA and ATAC appear to move freely in the nuclear environment, with little to no association with chromatin. We noted that the fast population of the ATAC subunit ZZZ3 had a lower than expected $MW_A$ (0.32 MDa instead of

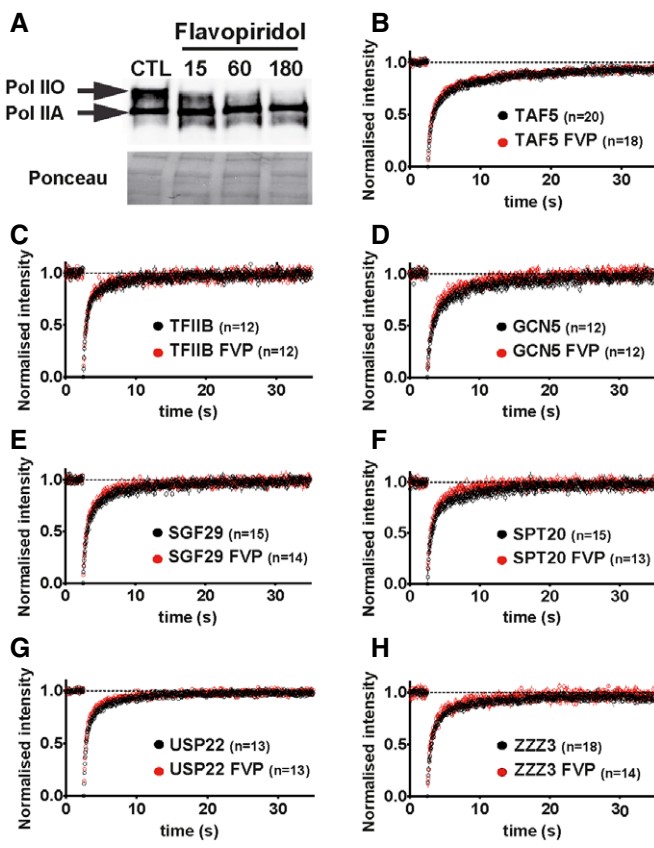

**Figure 3. FRAP analyses in control and flavopiridol-treated cell show no significant change in the dynamics of TAF5, TFIIB, SAGA and ATAC subunits.**

A    Cells were treated with 2 μM of flavopiridol (FVP) for the indicated time (in minutes). Upon FVP treatment, Pol II CTD phosphorylation levels were tested on whole cell extras of U2OS cells by Western blot analysis using an antibody raised against the CTD of RPB1. CTL: non-treated control cells. Hypo- (Pol IIA) and hyper-phosphorylated (Pol IIO) forms of RPB1 CTDs are labelled. Ponceau staining of the blot tested equal loading.

B–H    Comparison of average normalized FRAP curves obtained from control and FVP-treated cells corresponding to the following factors: (B) TAF5; (C) TFIIB; (D) GCN5 and (E) SGF29 (shared SAGA and ATAC subunits); (F) SPT20 and (G) USP22 (SAGA subunits); (H) ZZZ3 (ATAC subunit). Black dots: average FRAP curves derived from measurements in control cells. Red dots: average FRAP curves obtained in flavopiridol-treated cells. On the x-axis, time is represented in seconds (s). n: number of cell nuclei analysed for each factor.

Source data are available online for this figure.

~0.7 MDa). For ZZZ3, we also observed a significant overlap between the distributions of diffusion coefficients of its fast population with that of eGFP (Fig 5D). This suggests that a significant amount of ZZZ3-eGFP may not incorporate into ATAC holocomplexes, and/or may exist in other smaller complexes, perhaps subassemblies of ATAC. Similar $MW_A$ measurements for the *fast* GTF subpopulations gave 1.99 MDa for TAF5, consistent with the size of the biochemically purified TFIID complex (~1.5 MDa, when calculating the TFIID core subunits twice; Bieniossek *et al*, 2013), and 0.43 MDa for TFIIB. The latter is much larger than the expected 35 kDa size of this protein, implying that TFIIB interacts with other GTFs in the nucleoplasm, as has been previously suggested (Ossipow *et al*, 1995).

In contrast to the *fast* population, the apparent molecular weights of the *slow* subpopulations correspond to much bigger assemblies (in the order of $10^3$ MDa; Table 1), far larger than the size of any of the known complexes for these proteins. This slower population thus likely corresponds to complexes that are interacting transiently with chromatin. Overall, both averaging (FRAP/FLIP) and single-molecule (FCS) techniques indicate that SAGA and ATAC, but also TFIID and TFIIB, have similar high mobility characteristics; and although the complexes are not stably immobilized on chromatin, a significant fraction of their mobile pool exhibits a transient association with chromatin.

## FCS reveals that coactivators, SAGA and ATAC, and GTFs, TFIID and TFIIB, exhibit reduced chromatin interactions upon inhibition of transcription elongation

To investigate with higher sensitivity, the effects of transcription inhibition on the *fast* and *slow* diffusing components of the co-activator complexes and PIC components, we performed FCS measurements on cells treated with FVP and compared the obtained distributions of apparent diffusion constants with control cells. Importantly, the expression levels of proteins in control and FVP-treated cells were at similar range, and expression level variability did not affect the precision of diffusion constant measurement (Appendix Fig S4). As for control cells, FVP-treated cells had bi-modal distributions of apparent diffusion constants (Appendix Table S1), but in most cases the fraction of *slow* component was reduced and the fraction of *fast* component was increased (Fig 6A–F). In some cases, the peak of the *slow* component was shifted towards higher-mobility values. Altogether, this suggests that when Pol II is not actively transcribing, SAGA and ATAC, as well as TAF5 and TFIIB, exhibit reduced association with chromatin.

## FCS analysis underlines the key role of H3K4me3 for the recruitment of the slow, chromatin-associated population of SAGA and ATAC in living cells

SAGA and ATAC have subunits that can read histone post-translational modifications (PTMs). We hypothesized that such interactions may affect chromatin association and thus influence the mobility, or proportion of the chromatin-associated *slow* population of SAGA and ATAC. To test this hypothesis, we assessed whether FVP treatment induces changes in transcription-related histone marks, and whether such changes could be linked to the reduction in the chromatin-associated population of SAGA and ATAC. We prepared acidic extracts from FVP-treated cells and analysed the global abundance of major transcription-related histone modifications on bulk histones for up to 3 h upon addition of the drug (Fig 7A). As expected, we observed a quick decrease in H2Bub levels, since the deposition of the H2Bub1 mark is Pol II transcription elongation-dependent, coupled through RAD6/RNF20/RNF40 and PAF complexes (Kim *et al*, 2009). On the other hand, we did not detect changes in the levels of H3K36me3 or H3K9ac (a mark read and deposited by the GCN5), in agreement with previous observations (Tjeertes *et al*, 2009; Fuchs *et al*, 2014). In agreement with the cross-talk between H2Bub1 deposition and H3K4 trimethylation machineries (Thornton *et al*, 2014 and refs therein), we observed a reduction in

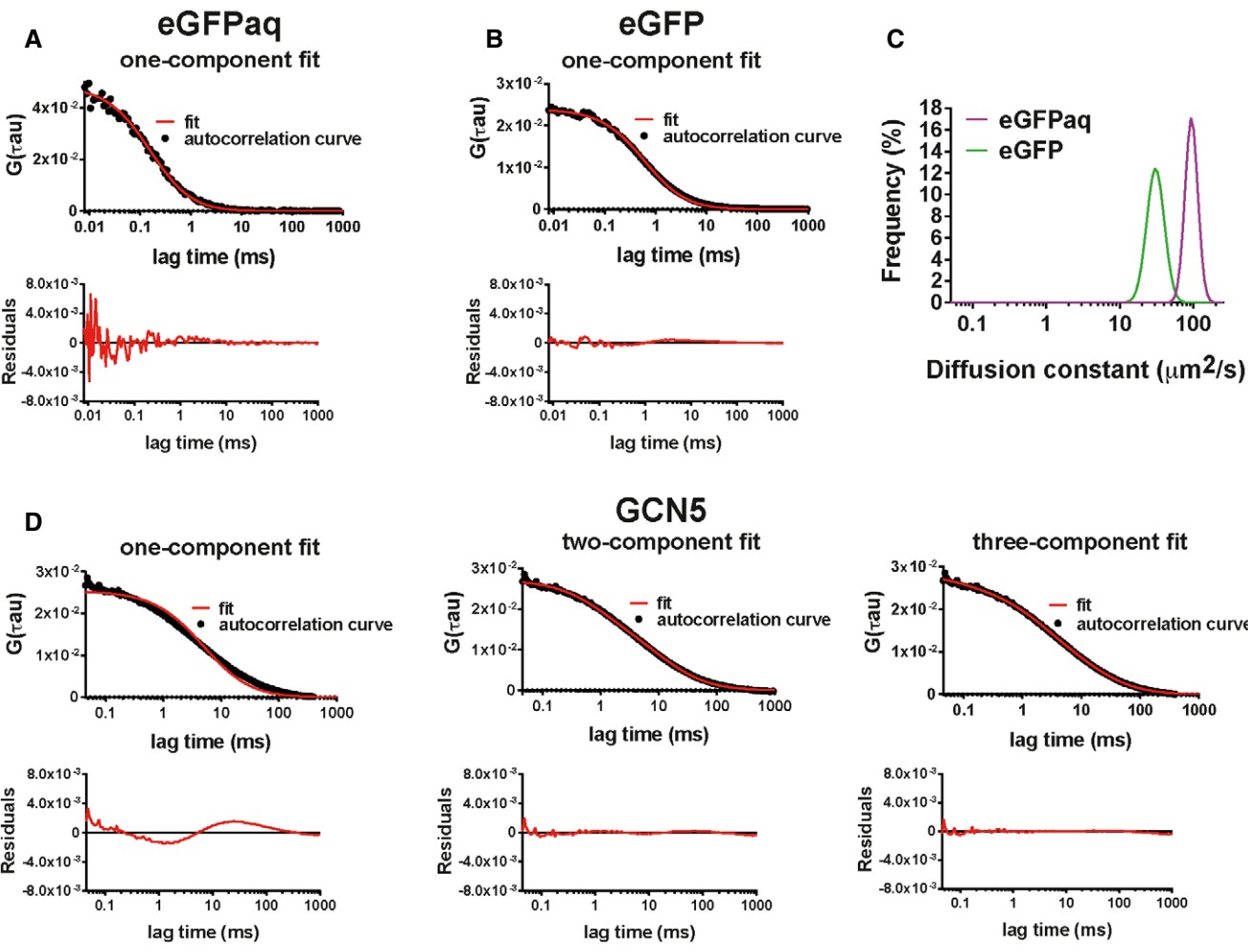

**Figure 4. Representative fittings of average autocorrelation curves with one or two-component free diffusion models.**

A   Average autocorrelation curve of purified eGFP diffusing in aqueous buffer (eGFPaq) is fitted with a one-component free diffusion model.
B   Average autocorrelation curve of free eGFP, from measurements obtained in the nuclei of U2OS cells, is fitted with a one-component free diffusion model.
C   Overlay of frequency distributions of apparent diffusion constants of eGFPaq and free eGFP obtained in the nuclear environment generated with the Maximum Entropy approach.
D   Comparison of the fit of the average autocorrelation curve obtained for eGFP-GCN5 in the nuclei of U2OS cells, with the one-component (upper left panel), two-component (upper middle panel) and three-component free diffusion models (upper right panel).

Data information: For each autocorrelation curve, the exactness of the fit with the respective model is shown by graphical analyses of the residuals (lower panels in A, B and D). G(τ): autocorrelation function. On the *x*-axis, time is represented in milliseconds (ms).

the level of H3K4me3, a mark read by the tandem tudor domains of SGF29, present in both ATAC and SAGA.

To further analyse in live cells to what extent the H3K4me3-SGF29 tudor domain interactions can affect chromatin binding of SAGA and ATAC, we deleted the tandem tudor domains from the eGFP-SGF29 protein. This construct, SGF29_DEL, was transiently expressed in U2OS, and its diffusion properties were analysed by FCS. The dynamics of SGF29_DEL were different from wild-type SGF29, with a pronounced reduction of the *slow*, chromatin-associated population of the mutant protein (Fig 7B). This result demonstrates, for the first time in living cells, that the tandem tudor domains of SGF29 mediate SAGA and ATAC interactions with chromatin.

To elucidate further how the interplay between SGF29 and H3K4me3 may regulate the global recruitment and association of

ATAC and SAGA with chromatin, we tested the effects of reduced H3K4me3 levels on SGF29 dynamics. We used siRNA to knock down ASHL2, a core subunit of the SET1/MLL/COMPASS-like methyltransferase complexes (Shilatifard, 2012), causing efficient reduction in H3K4me3 levels (Fig 7C; see also Steward *et al*, 2006). We performed FCS for transiently expressed eGFP-SGF29 in cells previously transfected with siRNA against ASH2L, and observed significantly reduced chromatin interactions (Fig 7D). Interestingly, the reduction in H3K4me3 levels has very similar effects on SGF29 dynamics as the deletion of the tandem tudor domains (Fig 7E). Thus, by using a quantitative, live-cell readout, and two independent approaches, we demonstrate that H3K4me3 is important for the recruitment of SAGA and ATAC to chromatin in human cells.

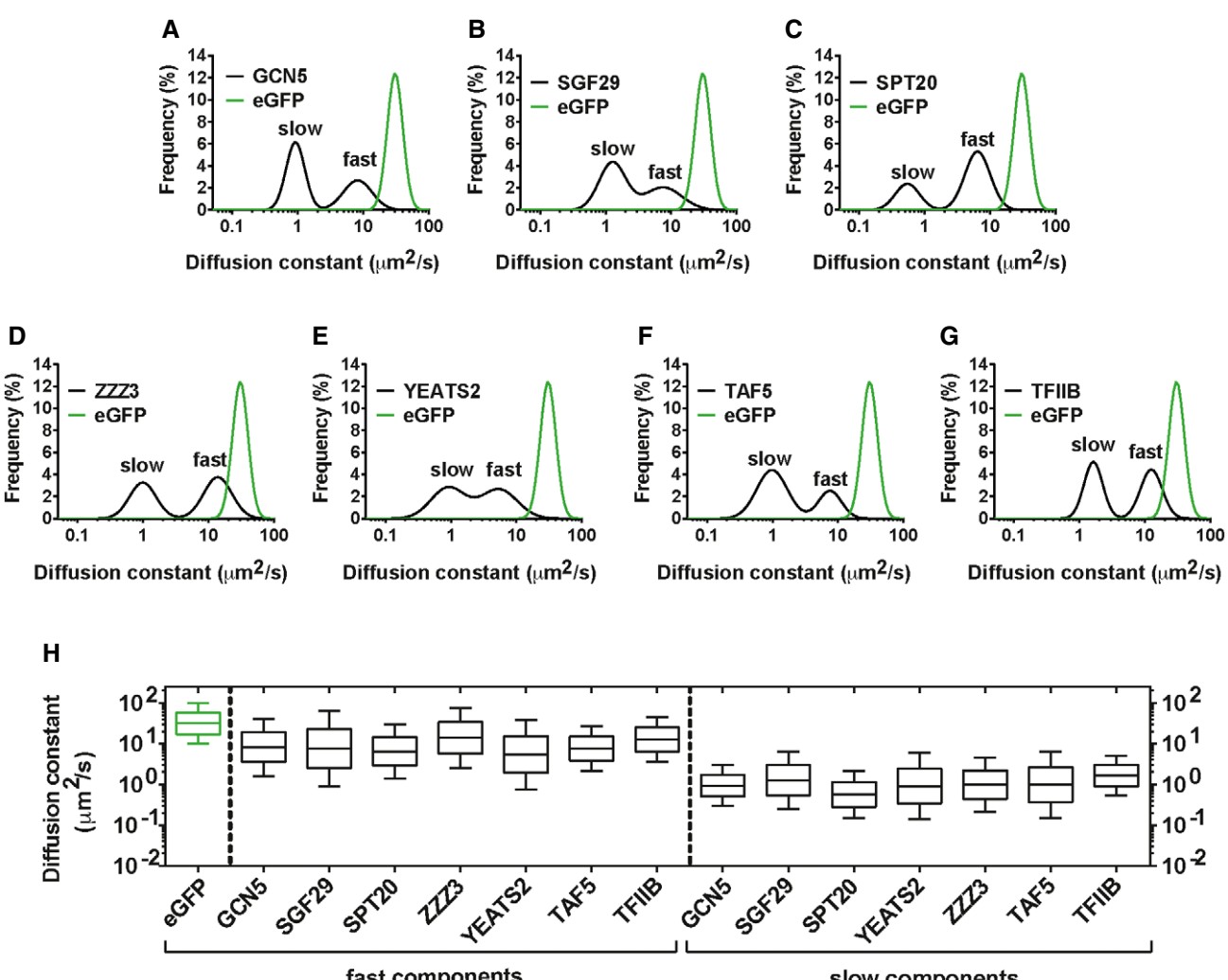

**Figure 5.  FCS measurements indicate that the mobile pool of SAGA and ATAC and PIC components, TAF5 and TFIIB, is dominated by a fast and a slow diffusing populations.**

A–G   Bi-modal distributions of apparent diffusion constants (*D*s) of eGFP-tagged proteins generated with the Maximum Entropy approach. In all frequency plots, the black line corresponds to the distribution of *D*s of the indicated factor. In all graphs for direct comparison with distribution of the *D*s of the respective eGFP fusion protein, the distribution of *D*s of free eGFP (giving a single peak) is included (green line). The peaks corresponding to *fast* and *slow* diffusing species are indicated on each panel. Distribution of *D*s of the following factors is shown: (A and B) GCN5 and SGF29 (common subunits of SAGA and ATAC); (C) SPT20 (SAGA subunit); (D and E) ZZZ3 and YEATS2 (ATAC subunits); (F) TAF5 (TFIID subunit); and (G) TFIIB.

H      Box-and-Whisker plot illustrating the comparison between the apparent diffusion constant values of the *fast* and *slow* component of each tested factor (black boxes) and the apparent diffusion constant of eGFP (green box). Values included in the box plots were calculated with the Maximum Entropy approach. The box represents values from the upper quartile (25% of data grater than this value) to the lower quartile (25% of data less than this value). The median (middle of the dataset) is indicated. Whiskers indicate maximum gratest value (top) and minimum least value (low) excluding outliers. The values included in the Box-and-Whisker plots include the pool of at least 3 independent experiments for each factor.

# Discussion

## The highly dynamic SAGA and ATAC coactivators and the GTFs, TFIID and TFIIB, have a fast and a slow diffusing nuclear population

SAGA and ATAC have been studied mainly by genetic, biochemical and structural approaches. Numerous studies have investigated gene-specific functions of the complexes and genome-wide approaches have been applied to define their binding sites.

However, it has been shown that transcription-associated proteins exhibit a broad range of mobility in the nuclei of living cells, with the stability of transcription factor-chromatin interactions linked to function (van Royen *et al*, 2011), and this information was missing for SAGA and ATAC complexes. Our FRAP and FLIP analyses classified the two coactivators as highly mobile factors, with no detectable immobile fractions, suggesting transient chromatin associations for these complexes. This behaviour is very similar to the dynamic GTFs, TFIID and TFIIB, and distinct from Pol II, which has a significant immobile fraction forming stable interactions with chromatin

**Table 1.  Summary of FCS derived mobility parameters of SAGA, ATAC, TFIID subunits and TFIIB.**

| | Apparent diffusion constant (µm$^2$/s) | | | | MW$_A$ (MDa) | | | | |
| --- | --- | --- | --- | --- | --- | --- | --- | --- | --- |
| | $D_{fast}$ (median) | $D_{fast}$ (range) | $D_{slow}$ (median) | $D_{slow}$ (range) | MW$_{fast}$ (median) | MW$_{fast}$ (range) | MW$_{slow}$ (median) | MW$_{slow}$ (range) | *n* |
| eGFP | 31.25 | – | – | – | – | – | – | – | 994 |
| GCN5 | 8.19 | 13–4 | 0.94 | 1.4–0.5 | 1.55 | 0.38–13.3 | 1,027 | 311–6,835 | 910 |
| SGF29 | 7.53 | 17–2 | 1.26 | 2–0.7 | 2 | 0.17–106 | 422 | 106–2,490 | 1.078 |
| SGF29_DEL | 4.52 | 8–2 | 0.69 | 1.2–0.3 | 9.25 | 1.7–106 | 2,600 | 495–31,650 | 1.035 |
| SPT20 | 6.42 | 10–3 | 0.56 | 0.8–0.3 | 3.21 | 0.85–31.6 | 4,643 | 1,670–31,650 | 756 |
| ZZZ3 | 13.82 | 25–7 | 0.98 | 1.6–0.5 | 0.32 | 0.054–2.5 | 885 | 208–6,835 | 641 |
| YEATS2 | 5.4 | 11–2 | 0.90 | 2–0.4 | 5.42 | 0.64–106 | 1,141 | 106–13,350 | 824 |
| TAF5 | 7.54 | 12–4.5 | 0.98 | 1.7–0.4 | 1.99 | 0.49–9.3 | 904 | 174–13,350 | 840 |
| TFIIB | 12.56 | 19–7 | 1.63 | 2.5–1 | 0.43 | 0.12–2.5 | 195 | 54–855 | 605 |

Mobility parameters of free eGFP and eGFP-tagged factors derived from the analysis of all the measurements with the Maximum Entropy approach (median values). The "range values" were obtained by using the full width at half maximum associated to each peak of the bi-modal distribution. In each cell, FCS measurements were sequentially repeated, 15 × 5 s in one to two points per nucleus. Thus, *n* values represent ~15× the number of total selected points and ~30× the total number of cells.

(Kimura *et al*, 2002; Hieda *et al*, 2005). Consequently, we conclude that the recruitment and function of the complexes on their chromatin substrate do not depend on direct and stable interactions with transcribing Pol II. This finding is of particular interest for the SAGA complex, which is known to exhibit its DUB activity on the gene body of actively transcribed genes (Bonnet *et al*, 2014). Our observations suggest that the removal of the ubiquitin moiety from H2Bub1 does not require SAGA to travel along the gene body together with elongating Pol II. Our data rather suggest that SAGA acts on H2Bub1 in a stochastic manner driven by the fast diffusion of the complex in the nuclear environment. These observations are also important as they explain the limited number of high-confidence DNA binding sites obtained in anti-subunit ChIP-seq profiles (Krebs *et al*, 2011; Lenstra & Holstege, 2012; Venters & Pugh, 2013). Our results suggest that in the case of complexes, which interact transiently with chromatin, monitoring their genomic sites of action by ChIP may not reveal their complete functional range. In addition, the rapid fluorescence recovery/loss described in FRAP/FLIP experiments is consistent with our previous observation that the genome-wide DUB activity of SAGA can be performed in < 10 min and in the absence of active transcription (Bonnet *et al*, 2014). Only complexes with high mobility and transient interactions with their substrate could act globally in such a short time.

The single-molecule sensitivity of FCS allowed us to characterize quantitatively the mobility of SAGA and ATAC subunits in the nuclei of living cells. The bi-modal distribution of apparent diffusion constants obtained for all of the tested subunits illustrates well that the pool of SAGA and ATAC complexes is not homogenous, in terms of mobility characteristics, but is rather dominated by two distinct populations: a *fast* and a *slow*. Molecules present in the *fast* population exhibit no or very few chromatin interactions, as indicated by their apparent molecular weight, which corresponds to complexes freely diffusing in the nuclear environment (Fig 7F). The *slow* population represents complexes that are interacting with chromatin in more stable manner (Fig 7F). However, it should be noted that in FCS only diffusing objects contribute to the fluctuations of fluorescence in the observation volume during the total time of each

measurement (see Materials and Methods). Thus, the *slow* population obtained in our FCS experiments should not be interpreted as (or compared with) proteins that are stably bound (for dozens of seconds or several minutes) to chromatin, as identified by other imaging (e.g. FRAP/FLIP kinetics of RPB1), biochemical or ChIP approaches. The estimated apparent diffusion constant of the slow component (ranging from 0.56 to 1.26 µm$^2$/s) is higher than what has been observed for transcription activators with high affinity for specific binding site motifs on the DNA sequence [e.g. $D_{slow}$ of eGFP-RAR-α (retinoic acid receptor alpha) is 0.05–0.10 µm$^2$/s (Brazda *et al*, 2011)]. These observations together suggest that the "slow" population of histone-modifying coactivators associates with chromatin in a more dynamic manner than *bona fide* DNA sequence-specific binding transcription factors, such as nuclear receptors. Similarly, although the two GTFs, TFIID and TFIIB, have been described as highly mobile in human cell nuclei (Sprouse *et al*, 2008; de Graaf *et al*, 2010), our FCS results show that the diffusing pool of these factors can also be further dissected into two distinct populations, analogous to SAGA and ATAC. It should be noted that, according to our measurements, we would expect that the *fast* and the majority of the *slow* population of SAGA and ATAC coactivator complexes, as well as for the GTFs, would be present in the "non-bound" fraction of a ChIP experiment, due to the transient nature of their interactions with the nuclear environment. Such information has to be taken into consideration to design and interpret genome-wide binding studies, as they may represent only a snapshot of the dynamic protein-chromatin interactions that take place in the nuclear environment. Overall, our quantitative analysis supports the notion that SAGA and ATAC interact transiently with chromatin (Fig 7F).

## Imaging measurements suggest that SAGA and ATAC complexes exist in the nuclei as holocomplexes

It has been suggested that components of multisubunit transcription-associated complexes may have very different mobility characteristics that can be resolved, either in the nucleoplasm or on the

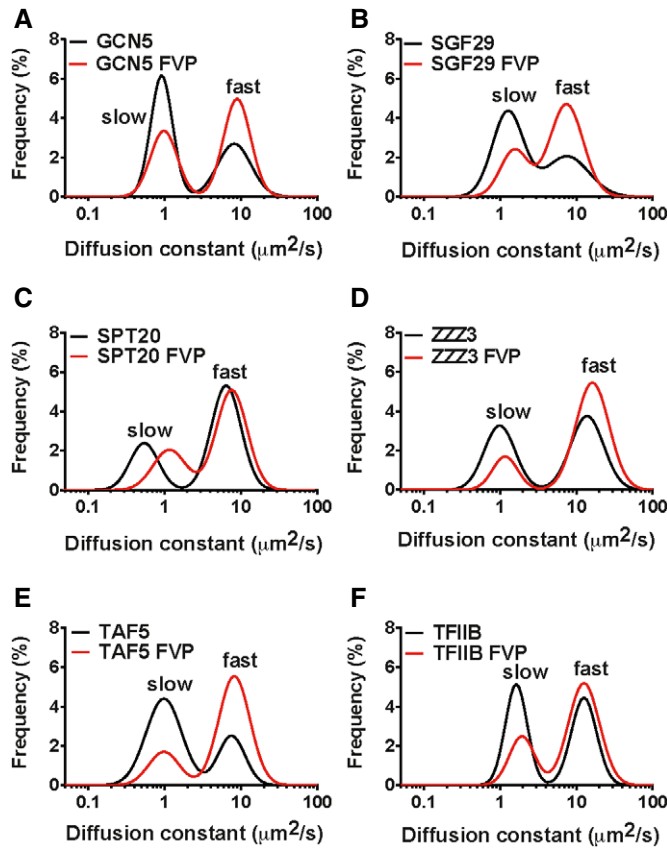

promoters where these complexes function. Such models have been put forward by measuring different recovery rates for distinct Pol I (Dundr *et al*, 2002), or Pol II (Sprouse *et al*, 2008) subunits, but also from observations regarding the inherent inefficiency of Pol I and II transcription (Darzacq *et al*, 2007; Gorski *et al*, 2008). On the contrary, other studies suggested that Pol II assembly occurs in the cytoplasm (Boulon *et al*, 2010; Czeko *et al*, 2011). With our FRAP/FLIP experiments, we did not detect any significant differences in the recovery rates among subunits of SAGA or among subunits of ATAC. In addition, comparison of FRAP/FLIP rates between SAGA- and ATAC-specific components showed that subunits of the two complexes behave in a very similar way. Our FCS observations are also in agreement with the FRAP/FLIP results, as we do not find significant differences between the *D*s of common or specific subunits of the complexes (Table 1). At present, we have no indications that SAGA or ATAC subunits have significantly different nuclear mobility characteristics (with the exception of ZZZ3), which would otherwise suggest the existence of multiple nuclear subcomplexes/modules that are sequentially recruited to their sites of action. Thus, similar to Pol II, SAGA and ATAC complexes may be assembled as holocomplexes in the cytoplasm and imported in the nucleus, where they exhibit their function. However, we cannot exclude the possibility that SAGA and/or ATAC subcomplexes would exist in the nucleus, but that their assembly into holocomplexes is too fast for their detection with the methods applied in the current study.

### Dynamics of SAGA and ATAC mobility during transcription

Our FRAP/FLIP experiments performed in normal conditions demonstrated that SAGA and ATAC mobility and recruitment is independent of interactions with elongating RNA polymerase II. However, since the two complexes have multiple subunits with chromatin-interacting domains, we also investigated whether the chromatin state defined by active transcription affects the dynamics of the complexes. FRAP and FLIP analysis after FVP-induced transcription elongation inhibition did not detect any significant changes for SAGA or ATAC subunits. On the contrary, FCS measurements performed under the same conditions revealed that indeed the chromatin-interacting fraction of both complexes (*slow* component) is

**Figure 6. FCS reveals that SAGA, ATAC and PIC components, TAF5 and TFIIB, exhibit reduced chromatin interactions upon inhibition of transcription elongation.**

A–F  Frequency plots illustrating the changes observed in the bi-modal distributions of apparent diffusion constants (*D*s) of eGFP-tagged proteins upon flavopiridol (FVP) treatment. Comparison of distributions of *D*s measured in control cells (black line) or in flavopiridol-treated cells (red line) of (A) GCN5 and (B) SGF29 (shared subunits of SAGA and ATAC); (C) SPT20 (SAGA subunit); (D) ZZZ3 (ATAC subunit); (E) TAF5 (TFIID subunit); and (F) TFIIB.

**Figure 7. FCS analysis supports the key role of H3K4me3 for SGF29-mediated SAGA and ATAC chromatin interactions.**

A  U2OS cells were treated with 2 μM of flavopiridol for the indicated time (in minutes). Equal amounts (1 μg) of protein from acidic extracts were loaded on gels. Ponceau staining of the blots tested equal protein loading. Histone H3, H3K4me3, H2Bub, H3K9ac and H3K36me3 levels were tested by Western blot analysis as indicated. CTL: non-treated control cell extract.
B  Comparison of the distribution of frequencies of diffusion constants (*D*s) of eGFP-tagged SGF29 (black line) and the mutant lacking the tandem tudor domains, SGF29_DEL (orange line).
C  U2OS cells were transfected with siRNAs targeting the histone methyltransferase ASH2L or with an siRNA against non-coding (NC) sequences for 48 h. Whole cell extracts were prepared and tested by Western blot analyses to evaluate the efficiency of ASH2L downregulation by siRNA and the effects on global H3K4me3 levels, as indicated. Ponceau staining of the blots tested equal amount of protein loading. Dotted line indicates that the blots were cut.
D  Comparison of the distribution of frequencies of *D*s of SGF29 in cells transfected with siRNA against ASH2L (SGF29_siASH2L, blue line) and in cells transfected with siNC (SGF29_siNC, dashed line).
E  Overlay of the distribution frequencies of *D*s measured for SGF29 (black line), SGF29_DEL (orange line), or SGF29_siASH2L (blue line).
F  Model illustrating how the fast and slow diffusing populations of SAGA/ATAC complexes are influenced by changes in histone modifications (i.e. H3K4me3). Left part: When transcription is active, the abundance of histone marks with which SAGA/ATAC subunits interact (e.g. SGF29-H3K4me3) and favour the association of the complexes with chromatin. As a result, a population of chromatin-interacting complexes can be detected in living cells as a slow diffusing fraction. Right part: When transcription elongation is inhibited, and/or the abundance of transcription-associated histone modifications is reduced (i.e. H3K4me3-ion), the chromatin-associated fraction of SAGA/ATAC complexes (slow component) is decreased. Thus, the majority of complexes become part of the "freely" diffusing population (fast component). Yellow dots: H3K4me3 mark. Arrows indicate interactions of complexes with the chromatin.

Source data are available online for this figure.

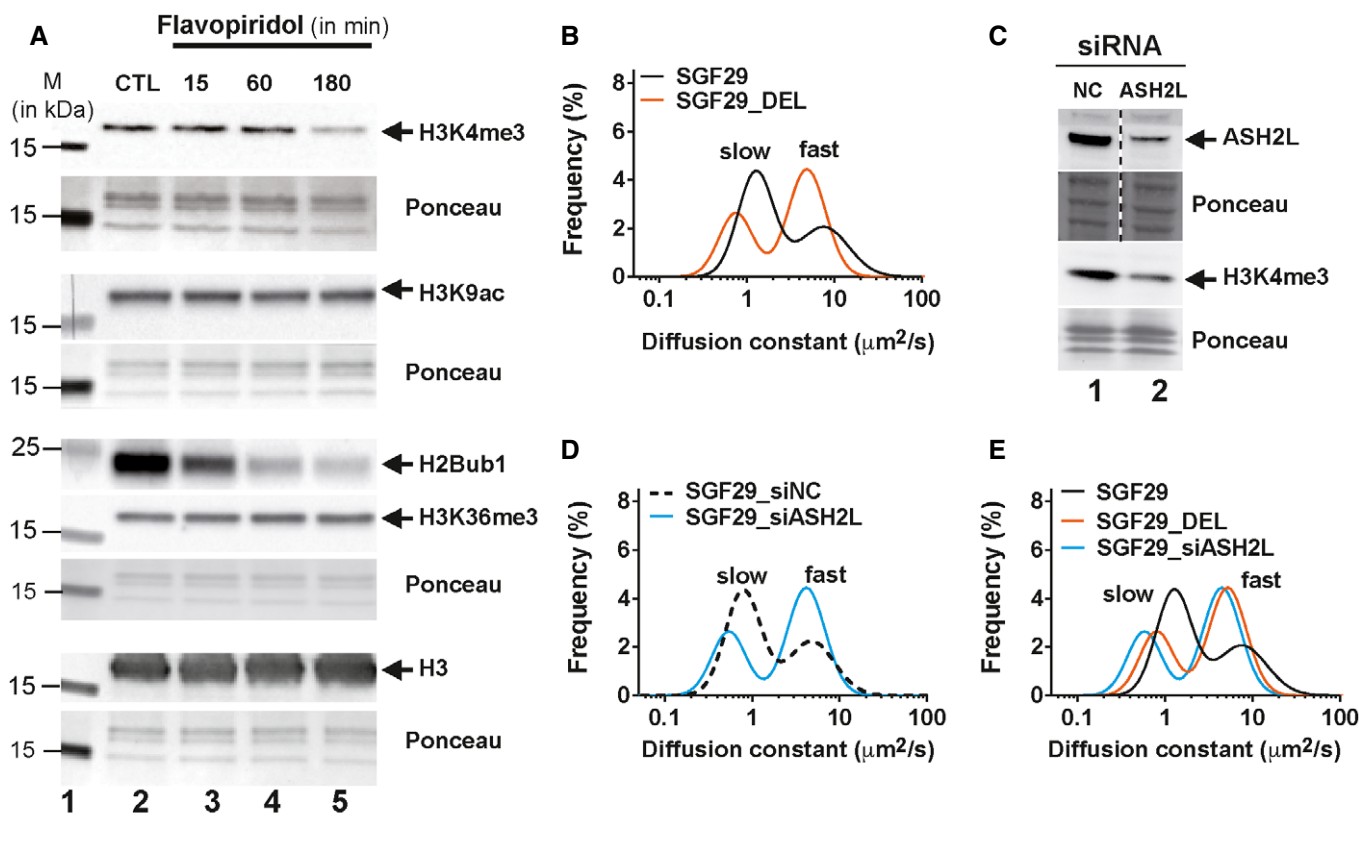

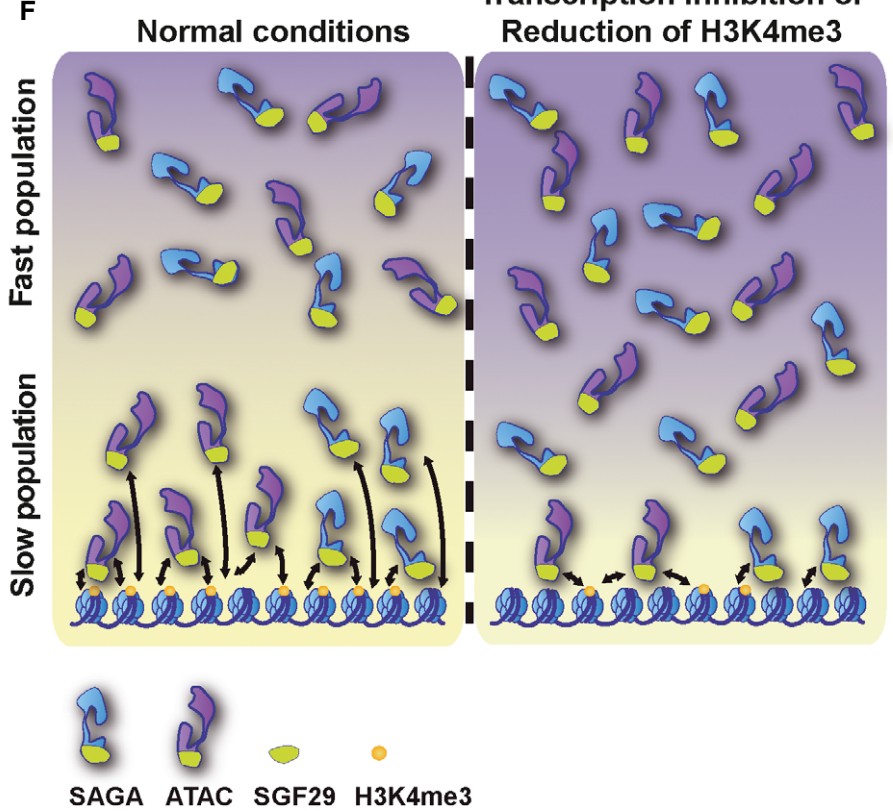

Figure 7.

reduced and/or exhibits reduced association with chromatin (i.e. SPT20). These results suggest that the chromatin status (i.e. histone PTMs, nucleosome composition/positioning) when transcription is blocked is less favourable for the recruitment of the two complexes. In addition, we demonstrated that although FRAP/FLIP did not detect significant changes in the mobility of TAF5 and TFIIB, the analysis of their diffusion properties by FCS revealed that these GTFs exhibit reduced interactions with chromatin upon inhibition of transcription elongation. Interestingly, our live-cell observations suggest that inhibition of Pol II elongation creates a chromatin environment which may increase the mobility of GTFs and coactivator complexes.

We further investigated this hypothesis by testing the levels of transcription-related histone modifications on bulk histones upon FVP treatment. The observed reduction in H3K4me3 levels prompted us to analyse the importance of the mark for the diffusion of SAGA and ATAC in living cells, both by deleting the tandem tudor domains of SGF29 and by reducing the global levels of H3K4me3 after down regulating the core subunit of HMT complexes. In both cases, we found a reduced association of SAGA and ATAC with chromatin, as indicated by the reduced proportion of the *slow* component of SGF29. The fact that we get similar, but not identical, FCS profiles for SGF29 with FVP treatment, siASH2L treatment or the tudor domain deletion (Fig 7E) suggests that additional mechanisms, other than the abundance of H3K4me3, may also affect the diffusion of SAGA and ATAC during these three different cellular conditions.

H3K4me3 is found around transcription start sites (TSSs) of transcribed and poised genes when analysed by ChIP-seq (Barski *et al*, 2007; Guenther *et al*, 2007). Particularly, it has been suggested that the extent to which H3K4me3 protrudes in the gene body may positively correlate with transcription activity of the respective gene and that certain histone marks, which correlate with elongating Pol II, are required for the deposition of H3K4me3 (Okitsu *et al*, 2010; Benayoun *et al*, 2014; Chen *et al*, 2015). Moreover, it has been hypothesized that variation in histone modifications may create alternative binding surfaces for potential histone tail readers, suggesting that histone modifications could function by creating molecular signalling gradients (Clouaire *et al*, 2014). In agreement, our data support a model in which SAGA and ATAC would associate more with chromatin when H3K4me3 is retained above a given threshold at a genome-wide level. It is expected that other reader domains of SAGA and/or ATAC subunits could also contribute to chromatin interactions. Upon transcription inhibition, the loss of these cumulative reader-histone mark interactions would contribute to the reduction in the described *slow* population. Nevertheless, the combination of our live-cell approaches confirms the importance of H3K4me3 mark for the global SAGA and ATAC recruitment and provides new evidence for the presence of a link between transcription-competent chromatin state and the recruitment of highly mobile complexes (Fig 7F). Interestingly, it has been shown in mouse embryonic stem cells that H3K9 acetylation, the mark known to be deposited by SAGA, is downstream of H3K4me3, pointing to a molecular function for H3K4me3 in targeting SAGA-dependent H3K9 acetylation (Clouaire *et al*, 2014). In conclusion, our results demonstrate that in the nuclear environment, there is an equilibrium between fast ("free") and slow ("chromatin-associated/modifying")

populations of SAGA and ATAC complexes that is regulated by the chromatin landscape, and which in turn is directly or indirectly dependent on active transcription.

# Materials and Methods

### Antibodies

The following primary antibodies at the indicated dilutions were used in this study:

Rabbit polyclonal anti-GFP (Abcam, ab290; dilution 1:2,000); mouse monoclonal anti-alpha-Tubulin (Sigma-Aldrich, T9026; dilution 1:1,000); rabbit polyclonal anti-ASH2L (gift from Winship Herr; dilution 1:2,000); rabbit polyclonal anti-Lamin A (Sigma-Aldrich, L1293; dilution 1:2,000); mouse monoclonal anti-RPB1 CTD (1PB 7C2, (Nguyen *et al*, 1996; dilution 1:20,000); rat monoclonal anti-RPB1 CTD phospho Ser2 (Active Motif, 61083; dilution 1:2,000); rabbit polyclonal anti-H3K4me3 (Abcam, ab8580; dilution 1:1,000); rabbit polyclonal anti-H3K36me3 (Abcam, ab9050; dilution 1:1,000); rabbit polyclonal anti-H3K9ac (Abcam, ab4441; dilution 1:1,000); rabbit monoclonal anti-H2Bub (Cell Signaling Technology, 5546; dilution 1:5,000); rabbit polyclonal anti-H3 (Abcam, ab1791; dilution 1:2,000), mouse monoclonal anti-GCN5 2GC 2C11 (dilution 1:1,000; Brand *et al*, 2001) or 5GC 2AG (dilution 1:1,000; Krebs *et al*, 2010); rabbit polyclonal anti-ZZZ3 2616 and anti-mADA3 2678 (dilution 1:1,000; Nagy *et al*, 2010); rabbit polyclonal anti-SPT20 3006 (dilution 1:1,000; Krebs *et al*, 2011); mouse monoclonal anti-TRRAP 2TRR 2D5 dilution (1:1,000; Robert *et al*, 2006), rabbit polyclonal anti-SGF29 2461 dilution (1:1,000) and rabbit polyclonal anti-YEATS2 2783 (dilution 1:1,000; Nagy *et al*, 2010).

### Transcription inhibition

To inhibit transcription elongation, cells were incubated with 2 μM flavopiridol (Sigma-Aldrich, #F3055). Evaluation of transcription inhibition was tested by Western blot (WB) assays on whole cell extracts of U2OS cells using an antibody against the CTD of RPB1 (1PB 7C2). For imaging experiments (FRAP, FLIP, FCS), drug was added ~2 h before imaging. Each treated sample was then used for 1–2 h. To ensure homogenous distribution of the drug, the required amount was first diluted in 500 μl of medium, prewarmed at 37°C, which was vortexed and finally added on cells growing on Ibidi μ (35 mm) high glass bottom dishes.

### Microscopy

#### FRAP and FLIP

For all live-cell imaging experiments in this study, U2OS cells were seeded on Ibidi μ (35 mm) high glass bottom dishes in phenol red free medium. FRAP was performed using a Nikon Ti-E inverted microscope (with Perfect Focus System, PFS) equipped with a CSU X1 Yokogawa confocal spinning disc and a 60 × CFI Plan Apo VC oil NA 1.40 (Nikon) objective. Fluorescence signal was detected with a Photometrics Evolve 512 EMCCD camera. During all FRAP measurements, cell was maintained at 37°C in 5% $CO_2$ using the Tokai Hit INUBG2E-TIZ stage top incubator. Roper iLas2 FRAP unit was used to bleach a circular nuclear ROI of 6 pixels radius,

corresponding to ~21.9 μm². The selected ROI was monitored every 50 ms. After 5 s (100 prebleach collected frames), the circular ROI was quickly bleached at maximum laser power with a double pulse of 50 ms. Subsequently, recovery of fluorescence was monitored with the same frequency for a total of 100–200 s. MetaMorph (Molecular Devices) image analysis software was used to process the raw images and obtain the average intensity of three ROIs, which were used for further normalization: the bleached ROI, the whole nucleus ROI and a ROI corresponding to background signal. EasyFRAP software (Rapsomaniki *et al*, 2012) was used to normalize the obtained background-subtracted fluorescence values. With the latter software, the signal of the bleached ROI was normalized to the average value of the last 50 prebleach frames, which was set to 1. Next, average fluorescence intensity of the whole nucleus at each time point was used to correct for signal loss in this ROI resulting from observational photobleaching and fluorescence loss during photobleaching. The obtained curves were normalized according to the full-scale normalization method (Ellenberg *et al*, 1997), in which the value of fluorescence intensity in the bleached ROI for the first post-bleach frame is set to zero.

FLIP was performed using the same hardware set-up as for FRAP. For the experiments described in this manuscript, a circular nuclear ROI of 17.5-pixel radius, corresponding to ~46.6 μm², was bleached every 3 s at maximum laser power by a double bleach pulse for a minimum total period of 4 min. Like in FRAP, raw images were quantified by MetaMorph software. Average fluorescence values were extracted from three ROIs required for FLIP data normalization: the first corresponds to the whole nucleus, the second corresponds to a neighbouring cell that was always included to the field of view (used to normalize for observational photobleaching), and a third region corresponds to background signal. Normalization of FLIP data was done according to the formulas described in Nissim-Rafinia and Meshorer (2011). In FLIP, the average corrected value of five prebleach frames of the whole nucleus region was normalized to 1.

### Fluorescence Correlation Spectroscopy (FCS)

FCS measurements were performed on a home-build two-photon system set-up based on an Olympus IX70 inverted microscope with an Olympus 60 × 1.2NA water immersion objective as previously described (Azoulay *et al*, 2003; Clamme *et al*, 2003), at 37°C. Two-photon excitation at 930 nm was provided by a tunable femtosecond laser (Insight DeepSee, Spectra Physics), and photons were collected using a set of two filters: a two-photon short pass filter with a cut-off wavelength of 680 nm (F75-680, AHF, Germany), and a band-pass filter of 520 ± 17 nm (F37-520, AHF, Germany). The fluorescence was directed to a fibre-coupled APD (SPCM-AQR-14-FC, Perkin Elmer) connected to a hardware correlator (ALV5000, ALV GmbH, Germany) allowing online calculation of the normalized autocorrelation function. In living cells, there is no real steady state for the fluorescence intensity fluctuations. For this reason, FCS measurements were sequentially repeated, typically 15 × 5 s with an excitation power adjusted to avoid photobleaching of the fluorophore within the excitation volume. Due to the inherent heterogeneity of the cellular medium, it is difficult to use a model with a well-defined number of diffusing species. To make no assumptions about this number, curves were first fitted by using a model that relies on the Maximum Entropy approach (Sengupta *et al*, 2003). In

this case, autocorrelation curves were analysed assuming a quasi-continuous distribution of diffusing components ($D_i$ with $i \gg 1$). The algorithm provides as an output the distribution of the obtained apparent diffusion constants for a specific condition. To test the robustness of the algorithm, we used free eGFP diffusing in aqueous solution, for which we obtained a single peak distribution centred around 92 μm²/s, a value which is in agreement with previous studies (Potma *et al*, 2001).

The Maximum Entropy fitting also resulted in a single peak distribution for free eGFP diffusing in the nucleus of UO2S cells. For all the other factors studied in this work, bi-modal distributions were obtained, demonstrating that the diffusion process was mainly dominated by the presence of two diffusing species with two different molecular weights. Thus, to determine in a quantitative manner the respective fraction of the two species, individual autocorrelation curves were fitted with the following model:

$$G(\tau) = \frac{1}{N}\left[(1-F)\left(\frac{1}{1+\frac{\tau}{\tau_{D1}}}\right)\left(\frac{1}{1+\frac{\tau}{S^2\tau_{D1}}}\right)^{\frac{1}{2}} + F\left(\frac{1}{1+\frac{\tau}{\tau_{D2}}}\right)\left(\frac{1}{1+\frac{\tau}{S^2\tau_{D2}}}\right)^{\frac{1}{2}}\right]$$

where $N$ is the average number of fluorescent species in the focal volume, $\tau$ the lag time, $\tau_{D1}$ and $\tau_{D2}$ the average residence time in the focal volume of the two diffusing species, $F$ the fraction of the high molecular weight diffusing species and $S$ a structural parameter defined as the ratio between the axial and lateral radii of the beam waist. If $F = 0$, the equation will account for the case where a single species diffuses within the observation volume.

The two-population fit was further used to test whether the measured parameters were showing any correlation with the expression level of cells used for our measurements. Particularly, to work in conditions where the diffusion time is independent of the expression level in the cells of interest, a threshold was used to filter out autocorrelation curves giving a number of molecules ($N$) above 200 in the observation volume of a given cell. The remaining cells, for which $N < 200$ in the observation volume, were re-analysed with the Maximum Entropy model to obtain the diffusion constant distributions.

In the frame of the Stokes-Einstein formalism, the diffusion constant of a sphere is related to its hydrodynamic radius according to:

$$D = \frac{k_B T}{6\pi\eta r_H}$$

where $k_B$ is the Boltzmann constant, T the absolute temperature in $K$, $\eta$ the viscosity and $r_H$ the hydrodynamic radius of the sphere. In this case, the diffusion constant scales with the molecular weight of the sphere at the power one-third. By taking as a reference the molecular weight of the free eGFP ($MW_{eGFP} = 27$ kD), it is possible to determine the apparent molecular mass (MW) of the labelled protein by using the following relation:

$$MW = MW_{eGFP}\left(\frac{D_{eGFP}}{D}\right)^3$$

where $D_{eGFP}$ and $D$ are the diffusion constants of the free eGFP and the eGFP-tagged protein, respectively.

Additional Materials and Methods are described in the Appendix Supplementary Methods.

**Expanded View** for this article is available online.

## Acknowledgements

We are grateful to M. HT. Timmers for TAF5 expression vector, to W. Herr for the anti-ASH2L antibody, to M. Vigneron for GFP-RPB1 and to present and former members of the Tora lab for cDNA constructs. We thank S. Bour for the illustration, J. Laporte for support for PK, F. El Saafin, D. Devys, S. Vincent and T. Sexton for critically reading the manuscript and for helpful comments, the IGBMC Imaging platform and the cell culture service for help and technical advice. N.V. was supported by an IGBMC PhD fellowship and a Fondation pour la Recherche Médicale (FRM) fellowship. This work was supported by funds from CNRS, INSERM, Strasbourg University, the EC Marie Curie-ITN (NR-NET) and the Agence Nationale de la Recherche (ANR-11-BSV5-010-02 Chromact; ANR-13-BSV6-0001-02 COREAC; ANR-13-BSV8-0021-03 DiscoverIID). This study was also supported by the European Research Council (ERC) Advanced grant (ERC-2013-340551, Birtoaction, to LT) and grant ANR-10-LABX-0030-INRT, a French State fund managed by the Agence Nationale de la Recherche under the frame program Investissements d'Avenir ANR-10-IDEX-0002-02.

## Author contributions

NV and ES performed the molecular laboratory work, NV, MK and PD performed imaging experiments. NV, PD and LT designed the study; NV, MK, PK and PD analysed data. All authors contributed text and figure panels to the manuscript. NV, PD, YM and LT wrote the manuscript. All authors gave final approval for publication.

## Conflict of interest

The authors declare that they have no conflict of interest.

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
