## [Review Process File · The EMBO Journal]

Manuscript EMBO-2016-96035

Coactivators and general transcription factors have two distinct dynamic populations dependent on transcription

Nikolaos Vosnakis, Marc Koch, Ms. Elisabeth Scheer, Pascal Kessler, Yves Mély, Pascal Didier & László Tora

Corresponding author: László Tora, IGBMC

Review timeline:

Submission date:	07 November 2016
Editorial Decision:	06 January 2017
Additional correspondence:	12 January 2017
Additional correspondence:	02 February 2017
Revision received:	04 May 2017
Editorial Decision:	02 June 2017
Revision received:	08 June 2017
Accepted:	15 June 2017

Editor: Anne Nielsen

Transaction Report:

1st Editorial Decision

06 January 2017

Thank you for submitting your manuscript for consideration by The EMBO Journal and my apologies for the extended duration of the review period, brought on by the recent holidays. Your study has now been seen by three referees whose comments are shown below. As you will see, while the referees express some interest in the work and topic in principle, they disagree on the advance provided and overall do not offer strong support for publication in The EMBO Journal.

I will not repeat all their individual points of criticism here, but it becomes clear that while the referees overall appreciate the technical quality of your work refs #1 and #2 are not convinced that the novelty and advance provided is sufficient for them to support publication here. In addition, ref #1 brings up major technical issues that may be addressable in principle but would not ameliorate the overall concern about the advance provided. You will see that referee #3 is more positive about the scope of the study, but given the strong hesitation from the other two referees we have to come to the conclusion that we cannot offer to publish your manuscript.

Thank you in any case for the opportunity to consider this manuscript. I am sorry we cannot be more positive on this occasion, but we hope nevertheless that you will find our referees' comments helpful.

REFEREE REPORTS

Referee #1:

The study by Vosnakis and coworkers examines the mobility of coregulatory protein complexes involved in transcription regulation. Although there are certainly some interesting observations and at first glance the authors may be on to an interesting finding, the study is way too premature and suffers from at least three flaws. A major part of the paper is confirmatory (point 1 below), there are major issues with most of the new findings (points 2-9) in part due to over-interpretation. In addition, the paper is not clearly written (examples in points 10-19). The study is far below the high standard usually shown by this group.

1. The majority of the paper (Figures 1-5) is a repeat/confirmation of previous work. This shows that GTFs/global transcription factors are mobile and that transcriptionally engaged RNA polymerase II is not. It is nice to get such an extensive confirmation but this impacts interest in the study, since the amount of new findings is limited and not well worked out. Perhaps it would have been possible to restrict this to two figures, or even start the study with the FCS work that reveals new phenomena.

There are major issues with the authors main conclusions (2-9, more or less in order of severity)

2. The authors conclude that their results demonstrate that the "equilibrium between fast (free) and slow (chromatin interacting) populations of SAGA and ATAC complexes is regulated by the active transcription-dependent chromatin landscape".

The results do not demonstrate a regulatory relationship. What they observe is that under conditions of decreased transcription there is a change in the proportion of fast and slow transcription complex populations. This does not mean that one is regulating the other.

3. The conclusion that a slow population of a complex is the chromatin interacting population is by inference of the molecular weight only. Furthermore, this inference is based on the exact same data/experiment that identifies a slow population in the first place. This conclusion (eg in the abstract, or pg 15, line 3, discussion) can not be made without additional (different) lines of experimental support to show the interaction.

4. The authors conclude that the H3K4me3 is an important histone mark for the (inferred) interaction of SAGA and ATAC with chromatin. Figure 9A shows what happens to the mark upon inhibition of transcription. Although the levels of H2Bub clearly decrease, the authors conclusion that H3K4me3 is decreased (and involved in the chromatin interaction) is flawed. Whereas there are major shifts between the slow and fast populations 60 minutes after addition of Flavopiridol, there is no observable effect on H3K4me3 at 60 minutes (Figure 9A, top). In the absence of a better, quantified assay for H3K4me3 levels, clearly showing significant drops in H3K4me3 at 60 minutes, this conclusion of the study is incorrect.

5. The authors need to explain/study the very different FCS profile of SPT20. It is a SAGA specific subunit and has a very different behaviour. FCS of a different SAGA specific subunit would be one possible starting point to understand this.

6. SGF29 shows a different behaviour in the Flavopiridol experiment versus the siRNA and tudor deletion experiment. In the first the slow population becomes faster and in the latter two the slow population becomes slower. This suggests that different things are happening in these experiments and should be explained.

7. There are some concerns about the accuracy of the MW determination. Two ATAC specific subunits that should be present in the same complex show > 10x difference in MW (!) The authors state that they "do not find significant differences between the Ds of common or specific subunits of the complexes (Table 1)". However no quantification is shown, nor it is not mentioned how this lack of significance was determined. A quantification of the differences of the MWs and Ds between the different subunits should be determined among the slow and fast diffusing populations. As the big difference in MW between some subunits (ZZZ3 and YEATS2, fast) suggest that these differences

could be significant.

8. The SGF29 mutant has different nuclear dynamics. This does not mean however (as the authors conclude) that the tandem tudor domains of SGF29 are mediating the interactions with chromatin. It would be nice to see this mutant incorporated in Table 1 as well.

9. It is not clear why RPB1 was not included in the FCS analysis. Would it not be interesting to see if Pol II also has 2 populations of different diffusing speed?

The manuscript is not very clearly written (especially the abstract and introduction) which leads to a lot of confusion (examples: 10-19).

10. Abstract: it is not clear what the introductory sentence has directly got to do with the findings. Are the authors suggesting that recruitment through activators is incorrect? If so, then please state this clearly. If not, change the introductory sentence.

11. The term active transcription-dependent chromatin landscape (end of abstract) is extremely vague and would be better replaced with a more concrete statement of what the authors actually mean.

12. Line 6, pg 3: it reads as if the authors are referring to histones as genetic elements. Moreover, histone modifications can work both ways, also making promoters/enhancers inaccessible for transcription. And to be even more precise, it is not the modifications themselves that are directly resulting in accessibility differences. The sentence after this one goes a little way in explaining such mechanisms, but overall, such vague, incomplete and partially incorrect introductory paragraphs can probably be better removed altogether.

13. Line 12, pg 4: "on the contrary" should probably be replaced by "Indeed". Otherwise, the sentence doesn't make sense.

14. Line 21, pg 4: perhaps the words "to take place" should be included after "suggested". Again, the sentence otherwise doesn't make sense.

15. Line 3, pg 5: This sentence too makes no sense: "However, as SAGA and ATAC interactions with chromatin is missing,". What do the authors mean? That there are no interactions. That no interactions have as yet been discovered/described? Or do they mean that it is difficult to detect such interactions? This part is central to the rationale of the experiments, and makes no sense!

16. Pg 7, 5 lines from bottom: the authors conclude that the GFP-tagged proteins are expressed at similar levels. Do the authors mean that each tagged protein is expressed at similar levels (correct) or that each individual cell expresses a similar amount of a given tagged protein (incorrect).

17. Figure S1B is not referred to in the main body of the text

18. Figures 3 and S1 seem to lack loading controls.

19. Page 13, 3rd paragraph the authors refer to figure 7I, however there is no figure 7I.

Referee #2:

In this manuscript by Vosnakis et al., the authors describe live-cell diffusion measurements on a host of transcription factors involved in the SAGA and ATAC complexes. Using FRAP, FLIP, and FCS, they observe two diffusive populations, fast and slow, and relate the slow population to chromatin binding with mutants and pharmacological inhibition. The data is of high quality, and the analysis is rigorous and thorough.

My primary concern is on the extent of the advance that is reported in this paper. We have known for some time that most factors exhibit multiple populations in the nucleus. For site specific factors,

these populations are usually due to specific and non-specific binding. As such, it comes as no particular surprise that SAGA and ATAC would show a diffusive population and a population which interacts transiently with chromatin. In that sense, the paper comes across as slightly behind the field in some regard. In the vanguard are studies which either: 1) use imaging of components at endogenous levels (achieve through gene editing or knock-in experiments) (PMID: 27855781), 2) utilize direct observation of single-molecules (i.e. PMID 25034201), or 3) couple the dynamic behavior with some downstream phenotype (PMID: 27015308). This paper doesn't rise to that level.

Nevertheless, the study is comprehensive in that many factors are examined using a unified set of protocols and analyses. Most of these factors have not previously been studied to my knowledge. Moreover, I find the mechanistic interpretation that GTFs show a slow population through binding to H3K4Me3 to be of general interest. In summary, the experiments are performed to a high standard, but the general principles revealed strike me as a modest advance.

Referee #3:

Report on manuscript "Coactivator complexes and PIC components have two dynamics populations dependent on active transcription", by N. Vosnakis et al.

This manuscript describes the dynamic behavior of two important transcriptional coactivator complexes, SAGA and ATAC, using a series of imaging techniques (FRAP, FLIP, FCS). This work stems from previous observation indicating that there is a poor correlation between steady-state binding of SAGA as measured by ChIP-Seq and change in expression levels upon depletion of SAGA subunits. This discrepancy has been explained by the Tora lab by proposing that binding of SAGA is dynamic and transient, and not well captured by ChIP. Indeed, mapping of the histone modifications deposited or removed by SAGA, together with additional experiments, showed that this complex functions on all genes (Bonnet et al., *Genes & Dev.* 2014, 28:1999-2012). However, all these evidences are indirect. The present study provides the first direct evidences that binding of SAGA and ATAC on chromatin is indeed transient and very rapid, and therefore represents an important advance.

By using Flavopiridol, a mutant SGF29 subunit and siRNAs against ASH2L, the authors further demonstrate that the binding events that they measure indeed occur on chromatin, and depend on both active transcription and the chromatin marks recognized by the "reader" subunits of the coactivator complexes.

In general, the data presented are solid and their interpretation is convincing. I thus fully support publication in EMBO Journal, provided the following minor points are addressed.

1-The FCS data in Table I are presented in terms of diffusion coefficient and apparent molecular weight. First, I would suggest to refer to the diffusion coefficient as an "apparent diffusion coefficient", since in the model of Figure 9, this value includes both true diffusion and binding. Second, and along the same lines, calculating an apparent molecular weight may not be the best way to interpret the slow component, as the slower diffusion is interpreted as transient binding to chromatin in Figure 9. Several models can be used to infer binding kinetics from FCS data, but in absence of direct binding measurements by single particle tracking (SPT), these can be misleading (Mazza et al. 2012, *NAR*, 40:e119). To keep things simple, I suggest that the authors could calculate a "mean chromatin residency time per diffusion volume", which may simply be the additional time that the slow particle takes to cross the FCS volume, assuming that they diffuse with D_{fast} and that they are chromatin-bound the rest of the time.

2-The FRAP experiments of Figure 1 are all displayed with a double normalization (pre-bleach to 1 and post-bleach to 0). It is essential that the authors also show in a Supplementary Figure the same data in a single normalization format (only to pre-bleached). Indeed, the post-bleached value carries important information especially for rapid molecules as is the case here, since the bleach depth decreases when rate of exchange of bleached molecules increases.

3-Regarding the experiment of the SG29 mutant subunit, the authors should show that this mutant subunit is incorporated into the coactivator complex as efficiently as the wild-type.

4-Regarding the experiment with ASH2L, it would be nice to add a control protein that does not bind this chromatin mark, to show that it is not affected by the siRNA (TAF5 or TFIIB ?).

5-Fits of the FCS data. To further reinforce the two-component model suggested by the Maximum Entropy approach, the authors should compare two-components and three-component fits to the auto-correlation function. A third component value of D similar to the others or representing a very small fraction of the population would provide a definitive support the two-component model. I also suggest that the authors indicate in Table I and also for the flavopiridol data the residuals of the two component fits, to make sure it works well in all cases.

6-In the introduction page 5, when the FRAP and FCS techniques are described, it would be better to give the time window that each technique can resolve, rather than the absolute value stated by the authors (milliseconds for FRAP and microseconds for FCS). This window depends on the microscopy set-ups and experimental design, but for FRAP is usually in the range of 0.05 to few hundred seconds, and about 100-1000 times less for FCS.

Along the same lines, the authors should precise throughout the text what time-scale they refer to for the "immobile" or "stably bound" molecules (e.g. immobile during how many seconds ?). These statements depend on the time-resolution of the technique used and readers unfamiliar with these techniques might have a hard time to get the meaning of these qualitative assessments (see for instance Discussion p20, the first three sentences of the page)

7-In Figure 3, it would be better to probe total H2B instead of H2Bub.

8-p7 I suggest to add (Boireau et al, J. Cell Biol, 2007, 179:291-304) to the citations of Darzacq, Shav-Tal 2007, Fromaget&Cook, 2007, etc...

9-p8, I suggest to indicate in the text the size of the FRAP ROIs used by the authors. In the following comparison with the previously published work of (de Graaf et al; Hielda et al.; Kimura et al.), I also suggest to include more details on what exactly is compared (I guess the $t_{1/2}$? but are these $t_{1/2}$ obtained using similar ROIs ?). This is important since the FRAP recoveries are highly dependent on the ROI size for diffusive molecules.

10-p13, I suggest to state in the text the range of diffusion coefficients measured for Dfast and Dslow.

11-p14, I disagree that "the observations that the values estimated by the in vivo FCS analyses for the fast populations of SAGA and ATAC subunits are in the MDa range indicate, first, that the tagged proteins incorporate in endogenous complexes and second, that this subpopulation of complexes moves in the nuclear environment freely, with no or very weak association with chromatin". Indeed, the fact that the Dfast correspond to the expected MW for the coactivator complexes could be coincidental, for instance if a significant fraction of the protein is not incorporated in the coactivator complex. What really shows proper incorporation are the IPs of Figure S1 and I would rephrase the sentence as follow: "the observations that the values estimated by the in vivo FCS analyses for the fast populations of SAGA and ATAC subunits are in the MDa range indicate that provided that the tagged proteins are efficiently incorporated in the endogenous complexes, then this subpopulation of complexes moves in the nuclear environment freely, with no or very weak association with chromatin". Likewise, I would delete the last sentence of this paragraph.

Finally, can the authors estimate from the IPs of Figure S1 how much of the protein is incorporated in the complexes and how much is free ? does the IP efficiencies observed correspond to what is seen for the endogenous proteins ? In fact, I would suggest to do the experiment the other way around: IP SAGA and ATAC using an endogenous untagged subunit and probe the blots using an antibody against the subunit that is tagged. This way, incorporation of GFP-tagged and untagged protein could be directly compared on the same blot/IPs. This would be an important control to reinforce the interpretation of the imaging data.

12-p21, with regard to the assembly of Pol II at the promoters (Sprouse et al., 2008), the authors

may want to cite contrary evidences (Czeko et al., Mol Cell 2011, 42-261-6; Boulon et al., Mol Cell 2010, 39:912-24;).

13-In Figure 7H, it is unclear to me of how the box plots are calculated (eg: where all the individual values come from). Is it by calculating Ds on a cell-to-cell basis ? Is it from a two component fit or from the maximum entropy approach ?

Additional correspondence - Authors

12 January 2017

Please find attached my rebuttal letter with potential answers to the Reviewers concerning our ms EMBOJ-2016-96035.

Based on our rebuttal letter, you will see that I consider that Reviewer 1 had a non-fair attitude when reviewing our manuscript and when judging the scientific advance.

Reviewer 2 seems to appreciate our work, but has non-justified subjective judgments (without really telling why) and Reviewer 3 is positive.

Note that such dynamic studies in live cells for chromatin remodeling coactivator complexes have never been done. In addition, the FCS studies for all the factors (GTFs and co-activators) are ground-breaking and thus, will be extremely interesting for the large transcription/chromatin community.

In conclusion, I would like to ask you to read our rebuttal letter (attached) and to reconsider your decision.

Thank you for your time, consideration and understanding.

Additional correspondence - Editor

02 February 2017

Thank you again for sending us a point-by-point response to the referee concerns for your manuscript. We have now consulted with an external expert advisor on both the technical concerns from ref #1 and on the overall scope of the study for The EMBO Journal. The outcome is that we have decided to invite you to submit a revised version of your manuscript (details included below).

For this revised manuscript I would ask you to focus your efforts on the following points:

-> While our arbitrating advisor found that the findings reported did offer sufficient advance to make your manuscript a strong candidate for publication in The EMBO Journal, this person also said the following:

'The paper contains a lot of data and some of the observations are interesting for the community, for example the conclusion that GTFs and co-activators have fast and slowly diffusing populations in the nucleus (although this may be expected I do not think it is all that clearly shown). However, the submitted text does not immediately convey the take-home message and does not immediately explain how the work goes beyond the state of the art. It is also too long and too descriptive; all the nebulous statements should be taken out. The compromised quality of the text apparently has led to a poor impression by referee #1.'

Since these comments on the manuscript format partly reflect those made by ref #1 I would strongly encourage you to move several of the FRAP/FLIP figures into the supplement. I realize that they are important controls for the following FCS assays on SAGA and ATAC but since they do not in themselves add much new insight their inclusion in the main manuscript is likely to muddle the overall message of the study.

In addition, I would ask you to provide a point-by-point response to the scientific/technical points

raised by all three referees with the revised manuscript.

Given the overall recommendations from our referees and the arbitrating advisor, I would thus like to invite you to submit a revised version of the manuscript, addressing the comments of all three reviewers. I should add that it is EMBO Journal policy to allow only a single round of revision, and acceptance or rejection of your manuscript will therefore depend on the completeness of your responses in this revised version.

Thank you for the opportunity to consider your work for publication. I look forward to your revision.

1st Revision - authors' response

04 May 2017

Referee #1:

The study by Vosnakis and coworkers examines the mobility of coregulatory protein complexes involved in transcription regulation. Although there are certainly some interesting observations and at first glance the authors may be on to an interesting finding, the study is way too premature and suffers from at least three flaws. A major part of the paper is confirmatory (point 1 below), there are major issues with most of the new findings (points 2-9) in part due to over-interpretation. In addition, the paper is not clearly written (examples in points 10-19). The study is far below the high standard usually shown by this group.

We were happy to learn that the Reviewer thought that we describe interesting findings. We answered and corrected the “three flaws” as described below.

1. The majority of the paper (Figures 1-5) is a repeat/confirmation of previous work. This shows that GTFs/global transcription factors are mobile and that transcriptionally engaged RNA polymerase II is not. It is nice to get such an extensive confirmation but this impacts interest in the study, since the amount of new findings is limited and not well worked out.

We respectfully disagree that all data presented in Figures 1-5 of the original version are “repeat/confirmation of previous work”. Figures 1-2 and 4-5 in the original version of our manuscript presented data that show novel FRAP and FLIP analyses with SAGA and ATAC subunits that were never done before. As we stated in the text, GFP-RPB1, GFP-TFIIB, GFP-TAF5 FRAP analyses and the GFP-RPB1 FLIP analysis were used i) as positive controls to demonstrate that our setups work correctly and ii) to compare the behaviour of our factors with that major players of Pol II transcription. However, we agree with the Reviewer that the photobleaching experiments with GFP-RPB1, the FRAP measurements concerning GFP-RPB1, GFP-TFIIB, GFP-TAF5 are important, but confirmatory experiments. Moreover, as the message of the FLIP measurements (without and with Flavopiridol, or FVP) basically confirms all the FRAP data, we agree with the Reviewer that these Figures may reduce the interest of the general reader in the main message. Thus, as required, all these important control experiments have now been transferred to Extended View Figures 1-3. Expanded View (EV) figures are those which are particular value to specialist readers, but which are not essential to follow the main thread of the paper for the general reader. In summary, we have transferred from original Figure 1 all the photobleaching panels concerning GFP-RPB1, and previous panels B, C and D to EV Figure 1 (see panels A-D), original Figure 2 containing all the FLIP experiments to EV Figure 2, panels B (eGFP +/- FVP) and C (RPB1 +/- FVP) from original Figure 4 to EV Figure 1 (see panels E and F), and original Figure 5 (FLIP experiments +/- FVP) to EV Figure 3. Moreover, we made a lot of effort to reduce all the confirmatory statements.

Figure 3 of the original version (Figure 2 in the revised version) is a cellular fractionation experiment. This is an independent method to verify our FRAP and FLIP findings on the dynamics of nuclear mobility, which further illustrates that SAGA and ATAC are not tightly associated to chromatin, which was not published before.

Perhaps it would have been possible to restrict this to two figures, or even start the study with the FCS work that reveals new phenomena.

As required, we have reduced the number of main Figures by two. In addition, we have alleviated original Figure 1 by four panels and original Figure 4 (presently Figure 3) by two panels. We also simplified the corresponding description of the data and shortened the manuscript by two pages.

There are major issues with the authors main conclusions (2-9, more or less in order of severity)

2. The authors conclude that their results demonstrate that the "equilibrium between fast (free) and slow (chromatin interacting) populations of SAGA and ATAC complexes is regulated by the active transcription-dependent chromatin landscape". The results do not demonstrate a regulatory relationship. What they observe is that under conditions of decreased transcription there is a change in the proportion of fast and slow transcription complex populations. This does not mean that one is regulating the other.

We have observed that under conditions where transcription elongation is inhibited there is a reduction in the fraction of the slow population of tested factors. These changes are very reproducible, for all the tested complexes and clearly different from a "random change" in the proportion" of the two populations. In addition, we have observed a shift from slow to fast populations when we inhibited histone H3K4 trimethylation by ASH2L siRNA (new Figure 7C; originally Fig. 9C). Thus, we have drawn our conclusion by summarizing the results based on two series of experiments, inhibition of transcription elongation and inhibition of histone H3K4 trimethylation. Nevertheless, as required, we have rephrased our conclusions to avoid giving the impression that "one is directly regulating the other" (see the new Abstract and end of page 20 and top of page 21 of the revised ms).

3. The conclusion that a slow population of a complex is the chromatin interacting population is by inference of the molecular weight only.

This conclusion has been reached on the basis of evidence, similarly as has been done by Brazda et al., *Mol Cell Biol.* 2014, 34(7):1234-45. Estimation of apparent molecular weight in FCS studies and its interpretation in a biological/cellular context is a standard for interpreting such type of data.

Furthermore, this inference is based on the exact same data/experiment that identifies a slow population in the first place. This conclusion (eg in the abstract, or pg 15, line 3, discussion) cannot be made without additional (different) lines of experimental support to show the interaction.

We have done several live cell FCS experiments to show that SAGA and ATAC are indeed interacting with the chromatin in live cells a) by deleting the H3K4me3-binding domain of SGF29, and b) by inhibiting H3K4me3-ityon and studying SGF29 (a common subunit of SAGA and ATAC) behaviour. The results of SGF29_DEL and SGF29 upon siASH2L, demonstrate clearly that when the abundance of a histone mark (H3K4me3) known to be important for the interaction of the SAGA/ATAC coactivators with chromatin is downregulated, the high molecular weight population ("slow") is reduced. These observations strongly support the characterisation of the "slow" population as chromatin interacting (see also point 8). Moreover, our very different biochemical cellular fractionation experiment also shows that a weak, but detectable fraction of SAGA and ATAC can associate with the chromatin under the cellular extraction conditions used (Figure 2). Nevertheless, as required, we have changed our conclusions on page 15 line 3, and we deleted "chromatin interacting" from the 11th line of the Abstract.

4. The authors conclude that the H3K4me3 is an important histone mark for the (inferred)

interaction of SAGA and ATAC with chromatin. Figure 9A shows what happens to the mark upon inhibition of transcription. Although the levels of H2Bub clearly decrease, the authors' conclusion that H3K4me3 is decreased (and involved in the chromatin interaction) is flawed. Whereas there are major shifts between the slow and fast populations 60 minutes after addition of Flavopiridol, there is no observable effect on H3K4me3 at 60 minutes (Figure 9A, top). In the absence of a better, quantified assay for H3K4me3 levels, clearly showing significant drops in H3K4me3 at 60 minutes, this conclusion of the study is incorrect.

We have now repeated the criticized western blot experiment (see new Figure 7A, previously 9A). The new experiments show an about 35% decrease in the H3K4me3-ion mark between 60- and 180-minute time-points, when carrying out western blots (WB) on acidic extracts obtained on 10^6 cells.

For the single cell live FCS experiments, cells were treated with flavopiridol (FVP) for 60 min, before they were transferred to the imaging facility for the FCS microscopy measurements. As a result, each FVP treated sample was used for microscopy during an additional period of 60-120 minutes (in a total of 120 to 180 min), depending on how easy was to find cells with the adequate low expression levels. Thus, we actually carried out the single cell measurements in the time interval when the reduction on H3K4me3 levels became obvious (by WB assay). We apologize for not being more precise in describing the timing of our experiments that is now corrected in the text (page 10), corresponding sections of the Figure legends and Materials and Methods of the revised version.

5. The authors need to explain/study the very different FCS profile of SPT20. It is a SAGA specific subunit and has a very different behaviour. FCS of a different SAGA specific subunit would be one possible starting point to understand this.

The profile of SPT20 results in two clearly distinct populations, slow and fast, as for every other subunit of SAGA (GCN5, SGF29). It is true that the estimated apparent molecular weight of the slow population of SPT20 is higher than that of the other subunits. This is something that we cannot interpret so far with our available data. From the technical point of view it is clear that we used the same protocol for all the studied factors. The observed difference has likely a biological meaning, however at this point we do not want to overstate our findings. We can assume that a portion of SPT20 that would integrate only in a building block of SAGA would interact with chromatin more permanently than other subunits of the complex. Another explanation could be that SPT20 also interacts with other complexes and/or structures, which we cannot identify with the applied "in vivo" methods. It is definitely an interesting observation which we will further investigate in the future. Focusing on new subunit(s) of the complexes (ATAC or SAGA) would be beyond the scope of this study, as we would need to repeat all the FRAP, FLIP and FCS analyses +/- FVP concerning the new subunit(s). This point has now been better described and discussed on pages 14 (top of the page) and 19.

6. SGF29 shows a different behaviour in the Flavopiridol experiment versus the siRNA and tudor deletion experiment. In the first the slow population becomes faster and in the latter two the slow population becomes slower. This suggests that different things are happening in these experiments and should be explained.

The fast and the slow populations of SGF29 show a similar behaviour upon Flavopiridol treatment, tudor domain deletion and ASH2L siRNA knock down experiments, namely in all of these three independent experiments we observe a global reduction of the slow population

and a mirroring increase in the population of the fast population (see new Figure 7B, and D, previously 9B and D). Thus, these FCS experiments do suggest that inhibition of transcription, inhibition of H3K4me3 binding of the complexes, or inhibition of H3K4me3- γ all reduce the slow, chromatin interacting fraction, and increase the proportion of free fast diffusing complexes in the nucleus. We have changed the colours of our graphs, to make our data clearer for non-specialized readers (new Fig. 7D and E).

However, we agree with the Reviewer that minor shifts can be observed in the maximum peaks of the slow populations. It has to be noted that the sensitivity of FCS may change slightly when fine tuning the equipment before each measurement and during the collection of data. Given the sensitivity of FCS, one would not expect to obtain identical profiles when analysing data resulting from totally independent treatments (i.e. siRNA and drug treatment). The fact that we do not get identical profiles suggests that potentially additional mechanisms, other than the abundance of H3K4me3, may also affect the diffusion of SGF29 (SAGA and/or ATAC) when cells are treated with FVP. Similarly, siASH2L most probably does not fully recapitulate all the effects of FVP in the nuclear environment. Still, the trend in the changes observed in the two conditions is strikingly similar and indicates a potential functional link between the two processes. Note however, that trying to provide a biological explanation to these “shifts” might be considered by the biophysics community as over-interpretation of the data, as long as no extended investigations (that are beyond the scope of this paper) will be performed. These points have now been better described on page 15 and pages 19 (bottom lines)-20 (top lines).

7. There are some concerns about the accuracy of the MW determination. Two ATAC specific subunits that should be present in the same complex show $> 10x$ difference in MW (!) The authors state that they "do not find significant differences between the Ds of common or specific subunits of the complexes (Table 1)". However no quantification is shown, nor it is not mentioned how this lack of significance was determined. A quantification of the differences of the MWs and Ds between the different subunits should be determined among the slow and fast diffusing populations. As the big difference in MW between some subunits (ZZZ3 and YEATS2, fast) suggest that these differences could be significant.

This observation is not related to the accuracy of the MW determination per se. As shown in the profile of ZZZ3, we have a higher overlap between the fast population distribution and that of eGFP. This is more evident than for any other tested subunit (e.g. YEATS2, another ATAC subunit). This fact is affecting the estimated MW based on ZZZ3 diffusion constant values. We assume that the observed overlap, which is highly reproducible, is either due to the presence of a free (non ATAC incorporated ZZZ3 population of molecules at endogenous conditions) or due to an imperfect incorporation of the eGFP-ZZZ3 in endogenous complexes. However, treatment with FVP shows reduction of the slow population of ZZZ3 suggesting that significant part of the eGFP tagged subunit is incorporated into functional ATAC. These points have now been better described and discussed on page 13.

8. The SGF29 mutant has different nuclear dynamics. This does not mean however (as the authors conclude) that the tandem tudor domains of SGF29 are mediating the interactions with chromatin. It would be nice to see this mutant incorporated in Table 1 as well.

We thank the Reviewer for this interesting suggestion. The required SGF29_{Del} mutant values have been included in revised Table 1 (see also point 10 of Reviewer 3).

9. It is not clear why RPB1 was not included in the FCS analysis. Would it not be interesting to see if Pol II also has 2 populations of different diffusing speed?

Classical FCS can mainly be used for measuring the diffusion properties of molecules that are highly mobile. However, as we and others have shown by FRAP and FLIP that a significant fraction of Pol II is immobile in the nucleus, the classical FCS method cannot be used to accurately measure Pol II behaviour in the nuclei of live cells. The presence of relatively immobile eGFP-RPB1 molecule (Pol II) in the FCS observation volume would result in increased photobleaching over the measurement time (5 seconds) that would not allow further reliable analyses. We apologize if this was not clearly explained in the original version of the manuscript. To perform FCS based measurement for partially immobile factors, such as Pol II, variations of FCS should be applied (e.g. scanning FCS) which is beyond the objective of our study.

The manuscript is not very clearly written (especially the abstract and introduction) which leads to a lot of confusion (examples: 10-19).

As required we have rewritten the Abstract, certain parts of the Introduction and in addition, our manuscript has been corrected and shortened (by two pages) by a native English speaker.

10. Abstract: it is not clear what the introductory sentence has directly got to do with the findings. Are the authors suggesting that recruitment through activators is incorrect? If so, then please state this clearly. If not, change the introductory sentence.

We apologize for the confusion. The introductory sentence has been changed.

11. The term active transcription-dependent chromatin landscape (end of abstract) is extremely vague and would be better replaced with a more concrete statement of what the authors actually mean.

As required, we deleted the criticized sentence and replaced it with more direct statements.

12. Line 6, pg 3: it reads as if the authors are referring to histones as genetic elements. Moreover, histone modifications can work both ways, also making promoters/enhancers inaccessible for transcription. And to be even more precise, it is not the modifications themselves that are directly resulting in accessibility differences. The sentence after this one goes a little way in explaining such mechanisms, but overall, such vague, incomplete and partially incorrect introductory paragraphs can probably be better removed altogether.

We apologize for the confusion. We have changed the criticized sentences and removed the introductory paragraph from the Introduction.

13. Line 12, pg 4: "on the contrary" should probably be replaced by "Indeed". Otherwise, the sentence doesn't make sense.

Corrected.

14. Line 21, pg 4: perhaps the words "to take place" should be included after "suggested". Again, the sentence otherwise doesn't make sense.

Corrected.

15. Line 3, pg 5: This sentence too makes no sense: "However, as SAGA and ATAC interactions with chromatin is missing," . What do the authors mean? That there are no interactions. That no interactions have as yet been discovered/described? Or do they mean that it is difficult to detect such interactions? This part is central to the rationale of the experiments, and makes no sense!

We apologize for the confusion. We corrected the sentence.

16. Pg 7, 5 lines from bottom: the authors conclude that the GFP-tagged proteins are expressed at similar levels. Do the authors mean that each tagged protein is expressed at similar levels (correct) or that each individual cell expresses a similar amount of a given tagged protein (incorrect).

We apologize for the confusion. We corrected the sentence. We meant that each tagged protein is expressed at similar levels.

17. Figure S1B is not referred to in the main body of the text

We apologize for the mistake that is now corrected.

18. Figures 3 and S1 seem to lack loading controls.

Figure 2 (previously Figure 3) is a cellular fractionation experiment, where we loaded 20 µg of protein from each fraction on the gel. Note that antibodies against α-Tubulin, Lamin A, histone H2Bub1 and histone H3 are all controls showing that the cellular fractionation was correctly carried out. We added an anti-H3 blot to this figure as a control for chromatin bound fraction, as also asked by Reviewer 3. As these are cytoplasmic, nuclear soluble and chromatin-associated protein fractions we could not find a protein marker that would be present equally in each fraction, thus we used equal amount of protein.

Appendix Figure 1A shows whole cell extracts (WCEs, 15 µg of each) prepared from cells transfected with the indicated plasmids. All the western blot panels were developed with an anti-GFP antibody. Figure 1B-G are 6 different anti-GFP immunoprecipitation (IP) experiments from the above described WCEs. Each blot contains the input (5%, 15 µg protein), the IP supernatant (SN, 5%, 15 µg of protein) and the IP eluate (33%) fractions.

We have now better described these experiments in the Figure legends of these panels.

19. Page 13, 3rd paragraph the authors refer to figure 7I, however there is no figure 7I.

We apologize for this error that is now corrected.

Referee #2:

In this manuscript by Vosnakis et al., the authors describe live-cell diffusion measurements on a host of transcription factors involved in the SAGA and ATAC complexes. Using FRAP, FLIP, and FCS, they observe two diffusive populations, fast and slow, and relate the slow population to chromatin binding with mutants and pharmacological inhibition. The data is of high quality, and the analysis is rigorous and thorough.

My primary concern is on the extent of the advance that is reported in this paper. We have known for some time that most factors exhibit multiple populations in the nucleus. For site specific factors, these populations are usually due to specific and non-specific binding. As such, it comes as no particular surprise that SAGA and ATAC would show a diffusive population and a population which interacts transiently with chromatin. In that sense, the paper comes across as slightly behind the field in some regard. In the vanguard are studies which either: 1) use imaging of components at endogenous levels (achieve through gene editing or knock-in experiments) (PMID: 27855781), 2) utilize direct observation of single-molecules (i.e. PMID 25034201), or 3) couple the dynamic behavior with some downstream phenotype (PMID: 27015308). This paper doesn't rise to that level.

Nevertheless, the study is comprehensive in that many factors are examined using a unified set of protocols and analyses. Most of these factors have not previously been studied to my knowledge. Moreover, I find the mechanistic interpretation that GTFs show a slow population through binding to H3K4Me3 to be of general interest. In summary, the experiments are performed to a high standard, but the general principles revealed strike me as a modest advance.

We were happy to learn that the Reviewer thought:

-that our manuscript was “comprehensive in that many factors are examined using a unified set of protocols and analyses;”

-that s/he found “the mechanistic interpretation that GTFs show a slow population through binding to H3K4Me3 to be of general interest”

-and that s/he found that “the experiments are performed to a high standard,”.

The Reviewer is suggesting that we should study 1) proteins at endogenous levels. As we stated in our manuscript at several places we made all the possible precautions to choose those single cells for our studies in which the fusion proteins were expressed at close to endogenous levels. S/he is also suggesting that we should observe 2) single molecules. We have to emphasize that “FCS is a method capable of measuring absolute molecular concentrations, diffusion rates, and molecular interaction dynamics in live cells (Magde et al. 1972). The measurement is based on observation of a single molecule or several molecules within a diffraction-limited spot in solution in a living cell.” The Reviewer’s suggestion that we should combine 3) our studies with phenotypic analyses is out of the scope of this manuscript.

Importantly Reviewer 2, in contrast to Reviewer 1, states that “Most of these factors have not previously been studied to my knowledge.”

Referee #3:

This manuscript describes the dynamic behavior of two important transcriptional coactivator complexes, SAGA and ATAC, using a series of imaging techniques (FRAP, FLIP, FCS). This work stems from previous observation indicating that there is a poor correlation between steady-state binding of SAGA as measured by ChIP-Seq and change in expression levels upon depletion of SAGA subunits. This discrepancies have been explained by the Tora lab by proposing that binding of SAGA is dynamic and transient, and not well captured by ChIP. Indeed, mapping of the histone modifications deposited or removed by SAGA, together with additional experiments, showed that this complex functions on all genes (Bonnet et al., *Genes & Dev.* 2014, 28:1999-2012). However, all these evidences are indirect. The present study provides the first direct evidences that binding of SAGA and ATAC on chromatin is indeed transient and very rapid, and therefore represents an important advance.

By using Flavopiridol, a mutant SGF29 subunit and siRNAs against ASH2L, the authors further demonstrate that the binding events that they measure indeed occur on chromatin, and depend on both active transcription and the chromatin marks recognized by the "reader" subunits of the coactivator complexes.

In general, the data presented are solid and their interpretation is convincing.

We thank Reviewer 3 for her/his positive comments and were happy to learn that s/he thought that the “data presented are solid and their interpretation is convincing, and that s/he fully supports publication of our work in EMBO Journal.”

I thus fully support publication in EMBO Journal, provided the following minor points are addressed.

We addressed these minor points as follows:

1-The FCS data in Table I are presented in terms of diffusion coefficient and apparent molecular weight. First, I would suggest to refer to the diffusion coefficient as an "apparent diffusion coefficient", since in the model of Figure 9, this value includes both true diffusion and binding. Second, and along the same lines, calculating an apparent molecular weight may not be the best way to interpret the slow component, as the slower diffusion is interpreted as transient binding to chromatin in Figure 9. Several models can be used to infer binding kinetics from FCS data, but in absence of direct binding measurements by single particle tracking (SPT), these can be misleading (Mazza et al. 2012, *NAR*, 40:e119). To keep things simple, I suggest that the authors could calculate a "mean chromatin residency time per diffusion volume", which may simply be the additional time that the slow particle takes to cross the FCS volume, assuming that they diffuse with D_{fast} and that they are chromatin-bound the rest of the time.

As required we have changed in Table 1 and in the manuscript diffusion coefficient (or constant) to "apparent diffusion coefficient" (or constant).

All the apparent diffusion constants can be converted to apparent diffusion times using the following equation: $\tau_D = \frac{\omega_0^2}{4D}$ where $\omega_0 = 0.34 \mu\text{m}$. It is thus possible to calculate the corresponding τ_{fast} and τ_{slow} . However, we do not think that the obtained values will give the “mean chromatin residency time”. Instead we think that they will indicate the average time needed, for the bound and free forms of the complexes, to cross the excitation volume. The interacting diffusing complex can be stable for minutes and only spend few milliseconds in the observation volume. We thus think that such calculations unfortunately will not be very

informative to describe “mean chromatin residency time”. However, if absolutely required we can put the τ_{fast} and τ_{slow} values in Table 1 (see also our answer to point 10 of Reviewer 3).

2-The FRAP experiments of Figure 1 are all displayed with a double normalization (pre-bleach to 1 and post-bleach to 0). It is essential that the authors also show in a Supplementary Figure the same data in a single normalization format (only to pre-bleached). Indeed, the post-bleached value carries important information especially for rapid molecules as is the case here, since the bleach depth decreases when rate of exchange of bleached molecules increases.

As required, the single normalization format is now also shown in new Appendix Fig. S3

3-Regarding the experiment of the SG29 mutant subunit, the authors should show that this mutant subunit is incorporated into the coactivator complex as efficiently as the wild-type.

As required, this co-IP experiment is now shown in new Appendix Fig. S1 panel G.

4-Regarding the experiment with ASH2L, it would be nice to add a control protein that does not bind this chromatin mark, to show that it is not affected by the siRNA (TAF5 or TFIIB ?).

The TFIID complex is supposed to bind H3K4me3 through its TAF3 subunit (Vermeulen et al. Cell. 2007 PMID:17884155). As TAF5 and TAF3 are both subunits of TFIID, the prediction would be that TAF5-GFP binding would be reduced by inhibition of H3K4me3 through ASH2L siRNA. Furthermore, as TFIIB is part of the PIC, of which the first step is the binding of TFIID to the promoter it is possible that TFIIB-GFP binding would also be reduced by the inhibition of H3K4me-ion. As these experiments may not give us a conclusive interpretation, we preferred not to carry them out.

5-Fits of the FCS data. To further reinforce the two-component model suggested by the Maximum Entropy approach, the authors should compare two-components and three-component fits to the auto-correlation function. A third component value of D similar to the others or representing a very small fraction of the population would provide a definitive support the two-component model. I also suggest that the authors indicate in Table I and also for the flavopiridol data the residuals of the two component fits, to make sure it works well in all cases.

As required, the three-component fit is now included in Figure 4 new panel B (Figure 6B in the original version). As indicated in the Results section the three-component fit did not significantly improve the goodness of the fit compared to the two-component model. Moreover, as required, reduced chi square values obtained from the two populations fits corresponding to all the tested factors +/- flavopiridol is now included in new Appendix Table 1. Together with the maximum entropy algorithm, Appendix Table 1 further supports the two populations model in all cases. This point is now better described on page 12 and 14.

6-In the introduction page 5, when the FRAP and FCS techniques are described, it would be better to give the time window that each technique can resolve, rather than the absolute value stated by the authors (milliseconds for FRAP and microseconds for FCS). This window depends on the microscopy set-ups and experimental design, but for FRAP is usually in the range of 0.05 to few hundred seconds, and about 100-1000 times less for FCS. Along the same lines, the authors should precise throughout the text what time-scale they refer to for the "immobile" or "stably bound" molecules (e.g. immobile during how many seconds ?). These statements depends on the time-resolution of the technique used and readers

unfamiliar with these techniques might have a hard time to get the meaning of these qualitative assessments (see for instance Discussion p20, the first three sentences of the page)

As required, we made references to time duration everywhere in the corresponding parts of the manuscript.

7-In Figure 3, it would be better to probe total H2B instead of H2Bub.

In the revised Figure 2, previously Fig. 3, we have now included an anti-histone H3 WB also to have a non-modified histone loading control, as required.

8-p7 I suggest to add (Boireau et al, J. Cell Biol, 2007, 179:291-304) to the citations of Darzacq, Shav-Tal 2007, Fromaget&Cook, 2007, etc...

The suggested references have been added.

9-p8, I suggest to indicate in the text the size of the FRAP ROIs used by the authors. In the following comparison with the previously published work of (de Graaf et al; Hielda et al.; Kimura et al.), I also suggest to include more details on what exactly is compared (I guess the $t_{1/2}$? but are these $t_{1/2}$ obtained using similar ROIs ?). This is important since the FRAP recoveries are highly dependent on the ROI size for diffusive molecules.

The size of the ROI is indicated in the Material and Methods (both for FRAP and FLIP experiments). As the reviewer correctly notices in the different studies different ROIs have been used depending on the microscopy set up/system used in each case. Thus, comparing $t_{1/2}$ between different set up could be misleading (as many parameters are different i.e. ROI size, sensitivity of detectors, frame rates and even expression levels).

That is the reason why our analysis for FRAP is semi-quantitative.

For example, in FRAP experiments, the similarity of results is based on the approximate time needed for each curve to reach full recover (approx. 10 s for SAGA/ATAC subunits). However, our main goal was to investigate and compare the presence (or absence) of immobile fraction within the selected time scales (i.e. 30-90 seconds for FRAP).

To make quantitative comparisons with previous studies we would have to calculate diffusion constant values from FRAP experiments. This would be really challenging given the fast diffusion observed for our factors of interest. Thus, for photobleaching experiments (FRAP/FLIP), we based our conclusions mainly on qualitative characteristic of fluorescence recovery/loss and we only use FCS to obtain quantitative information on the diffusion of our complexes of interest.

10-p13, I suggest to state in the text the range of diffusion coefficients measured for D_{fast} and D_{slow} .

As required, we show now also the “range” of diffusion coefficients measured for D_{fast} and D_{slow} in new Table 1.

11-p14, I disagree that "the observations that the values estimated by the in vivo FCS analyses for the fast populations of SAGA and ATAC subunits are in the MDa range indicate, first, that the tagged proteins incorporate in endogenous complexes and second, that this subpopulation of complexes moves in the nuclear environment freely, with no or very weak association with chromatin". Indeed, the fact that the D_{fast} correspond to the expected MW for the coactivator

complexes could be coincidental, for instance if a significant fraction of the protein is not incorporated in the coactivator complex. What really shows proper incorporation are the IPs of Figure S1 and I would rephrase the sentence as follow: "the observations that the values estimated by the in vivo FCS analyses for the fast populations of SAGA and ATAC subunits are in the MDa range indicate that provided that the tagged proteins are efficiently incorporated in the endogenous complexes, then this subpopulation of complexes moves in the nuclear environment freely, with no or vey weak association with chromatin". Likewise, I would delete the last sentence of this paragraph. Finally, can the authors estimate from the IPs of Figure S1 how much of the protein is incorporated in the complexes and how much is free ? does the IP efficiencies observed correspond to what is seen for the endogenous proteins ? In fact, I would suggest to do the experiment the other way around: IP SAGA and ATAC using an endogenous untagged subunit and probe the blots using an antibody against the subunit that is tagged. This way, incorporation of GFP-tagged and untagged protein could be directly compared on the same blot/IPs. This would be an important control to reinforce the interpretation of the imaging data.

The criticized sentences on page 12 (in the revised ms) have been rephrased and the "last sentence" was deleted.

12-p21, with regard to the assembly of Pol II at the promoters (Sprouse et al., 2008), the authors may want to cite contrary evidences (Czeko et al., Mol Cell 2011, 42:261-6; Boulon et al., Mol Cell 2010, 39:912-24;).

The suggested references have been added.

13-In Figure 7H, it is unclear to me of how the box plots are calculated (eg: where all the individual values come from). Is it by calculating Ds on a cell-to-cell basis ? Is it from a two component fit or from the maximum entropy approach ?

In revised Figure 5H (originally Fig 7H) the individual values are calculated on a cell-to-cell basis. They are derived from the two-component fit. All the individual traces for a given condition (more than 500 in general) have been used to determine the distribution of the diffusion constants using the maximum entropy approach. The box plots have been obtained from the individual peaks of the obtained distributions.

Thank you for submitting a revised version of your manuscript. It has now been seen by two of the original referees whose comments are shown below. In addition, I received informal comments from ref #1 who stated that the study has improved significantly and that the main message and general relevance are now clear. At the same time the referee mentioned to me that the overall readability of the study could be improved further and that the title could be more specific.

As you will see from this and the comments below, the referees now agree that all criticisms have been sufficiently addressed and recommend the manuscript for publication in The EMBO Journal. However, before we can go on to officially accept the manuscript there are a few editorial issues concerning text and figures that I need you to address in a final revision (in addition to the minor remaining points raised by the referees)

Thank you again for giving us the chance to consider your manuscript for The EMBO Journal, I look forward to your final revision.

 REFEREE REPORTS

Referee #2:

I did not have any specific technical concerns which needed to be addressed. In my last review, I stated: "the experiments are performed to a high standard, but the general principles revealed strike me as a modest advance." Since the results and conclusions are more or less the same as before, I think this statement still applies.

Referee #3:

Report on the revised manuscript of Vosnakis et al.

The authors have correctly answered the criticisms of the referees, and have done a good job in improving the presentation of the data and of their results.

A number of FRAP studies have been done with factors involved in transcription, but very dynamic factors are poorly resolved with this technique. The use of FCS to show that the rapid population of SAGA and ATAC can be resolved into freely diffusing and chromatin interacting (albeit very transiently) is important, as the demonstration that these interactions depend on active transcription and on interactions with H3K4me3.

SAGA and ATAC are key factors and their interactions with chromatin have not been studied before. This report further help to resolve conflicting results regarding their interaction with chromatin.

I have noted few typos:

-p7: a comma is missing "... for imaging experiments, they were weakly...."

-p7: references at the beginning of the second paragraph, #5287 should be removed.

-p19: last sentence of first paragraph is not clear.

-p20: first line, "... start sites... "

2nd Revision - authors' response

08 June 2017

The authors made the requested changes and submitted the final version of their manuscript.

3rd Editorial Decision

15 June 2017

Thank you for submitting the final revision, I am pleased to inform you that your manuscript has now been accepted for publication in the EMBO Journal.

If you have any questions, feel free to contact me. Thank you for your contribution to The EMBO Journal and congratulations on this nicely executed work!

Corresponding Author Name: Dr. Laszlo Tora

Journal Submitted to: The EMBO Journal

Manuscript Number: 96035R1